# Delivery Capacity and Anticancer Ability of the Berberine-Loaded Gold Nanoparticles to Promote the Apoptosis Effect in Breast Cancer

**DOI:** 10.3390/cancers13215317

**Published:** 2021-10-22

**Authors:** Chen-Feng Chiu, Ru-Huei Fu, Shan-hui Hsu, Yang-Hao (Alex) Yu, Shun-Fa Yang, Thomas Chang-Yao Tsao, Kai-Bo Chang, Chun-An Yeh, Cheng-Ming Tang, Sheng-Chu Huang, Huey-Shan Hung

**Affiliations:** 1Institute of Medicine, Chung Shan Medical University, Taichung 40201, Taiwan; bettychiu0603@gmail.com (C.-F.C.); ysf@csmu.edu.tw (S.-F.Y.); 2Division of Chest, Department of Internal Medicine, Feng Yuan Hospital, Ministry of Health and Welfare, Taichung 42055, Taiwan; 3Graduate Institute of Biomedical Science, China Medical University, Taichung 40402, Taiwan; rhfu@mail.cmu.edu.tw (R.-H.F.); kbwork2021@gmail.com (K.-B.C.); fireleafmaple@hotmail.com (C.-A.Y.); neoness0@gmail.com (S.-C.H.); 4Translational Medicine Research, China Medical University Hospital, Taichung 40402, Taiwan; 5Institute of Polymer Science and Engineering, National Taiwan University, Taipei 10617, Taiwan; shhsu@ntu.edu.tw; 6Changhua Hospital, Ministry of Health & Welfare, Changhua 51341, Taiwan; yuchest71@gmail.com; 7Department of Medical Research, Chung Shan Medical University Hospital, Taichung 40201, Taiwan; 8Division of Chest, Department of Internal Medicine, Chung Shan Medical University Hospital, Taichung 40201, Taiwan; his885889@gmail.com; 9School of Medicine, Chung Shan Medical University, Taichung 40201, Taiwan; 10Collage of Oral Medicine, Chung Shan Medical University, Taichung 40201, Taiwan; ranger@csmu.edu.tw

**Keywords:** berberine, collagen, gold nanoparticles, breast cancer, endocytosis

## Abstract

**Simple Summary:**

In this research, we aimed to evaluate the biological effects of physically gold nanoparticle-collagen nanocarrier incorporated with alkaloid berberine (Au-Col-BB) on non-transformed bovine aortic endothelial cells (BAEC) and Her-2 breast cancer cell lines through in vitro and in vivo assessments. Au-Col-BB showed better cytotoxicity, as well as significantly induced cell apoptosis in Her-2 cancer cells compared with normal cells (non-transformed BAEC). Further, Au-Col-BB also demonstrated better anti-cancer capacity for inhibiting cell growth in Her-2 tumor-bearing mice. In brief, we confirmed that the Au-Col-BB nanocarrier could be a potential nanodrug for increasing the efficiency of specific therapeutic effects in breast cancer disease.

**Abstract:**

Gold nanoparticles (AuNPs) were fabricated with biocompatible collagen (Col) and then conjugated with berberine (BB), denoted as Au-Col-BB, to investigate the endocytic mechanisms in Her-2 breast cancer cell line and in bovine aortic endothelial cells (BAEC). Owing to the superior biocompatibility, tunable physicochemical properties, and potential functionalization with biomolecules, AuNPs have been well studied as carriers of biomolecules for diseases and cancer therapeutics. Composites of AuNPs with biopolymer, such as fibronectin or Col, have been revealed to increase cell proliferation, migration, and differentiation. BB is a natural compound with impressive health benefits, such as lowering blood sugar and reducing weight. In addition, BB can inhibit cell proliferation by modulating cell cycle progress and autophagy, and induce cell apoptosis in vivo and in vitro. In the current research, BB was conjugated on the Col-AuNP composite (“Au-Col”). The UV-Visible spectroscopy and infrared spectroscopy confirmed the conjugation of BB on Au-Col. The particle size of the Au-Col-BB conjugate was about 227 nm, determined by dynamic light scattering. Furthermore, Au-Col-BB was less cytotoxic to BAEC vs. Her-2 cell line in terms of MTT assay and cell cycle behavior. Au-Col-BB, compared to Au-Col, showed greater cell uptake capacity and potential cellular transportation by BAEC and Her-2 using the fluorescence-conjugated Au-Col-BB. In addition, the clathrin-mediated endocytosis and cell autophagy seemed to be the favorite endocytic mechanism for the internalization of Au-Col-BB by BAEC and Her-2. Au-Col-BB significantly inhibited cell migration in Her-2, but not in BAEC. Moreover, apoptotic cascade proteins, such as Bax and p21, were expressed in Her-2 after the treatment of Au-Col-BB. The tumor suppression was examined in a model of xenograft mice treated with Au-Col-BB nanovehicles. Results demonstrated that the tumor weight was remarkably reduced by the treatment of Au-Col-BB. Altogether, the promising findings of Au-Col-BB nanocarrier on Her-2 breast cancer cell line suggest that Au-Col-BB may be a good candidate of anticancer drug for the treatment of human breast cancer.

## 1. Introduction

Cancer diseases are leading reasons of death around the world. Cancer cells trigger apoptosis resistance, metastasis, inflammation, and the breakdown of intercellular communication that cause poor immune response. The treatments of cancer include several clinical therapies, such as surgery, radiation, and chemotherapy. Although chemotherapy is an effective procedure to decrease the volume of primary tumor before surgery, a long-term chemotherapy treatment can cause various side effects, such as nausea, vomiting, fatigue, hair loss, leukopenia, mucositis, neurosensory disorders, and taste alterations in cancer patients, and induce multidrug resistance [1,2,3,4]. In the same way, there are still various bottlenecks that need to be conquered urgently. For instance, new drug development for cancer diseases requires plenty of time and costs expensive [5]. A literature figured out “Drug repurposing” to be a clinical strategy through the use of approved drugs for breast cancer therapeutics [5], such as the combination of insulin like growth factor 1 receptor (IGF1R) inhibitors with approved drug Rapamycin [6] demonstrates the inhibition of breast cancer cells proliferation [7].

Breast cancer is a commonly occurring cancer in human worldwide [8]. Her-2, the human epidermal growth factor receptors type 2, is a membrane receptor tyrosine kinase (RTK) which enhances cell proliferation and also overexpressed in various breast cancers. Her-2 oncogene is a member of human epidermal growth factor receptor family which located on chromosome 17q12 [9]. The Her-2 overexpressed breast cancer tends to grow faster and metastasize. Various clinical therapeutic drugs such as Herceptin (traztuzumab) are effective to target Her-2. Despite the significant progress in the field of nanomedicine, targeting Her-2 was still a barrier due to the genetic drift that could hide many important epitopes [10]. Thus, to conquer chemoresistance in breast cancer, using combo nanomedicine becomes a promising technique for chemotherapeutics. However, tumor accumulation of nanocarriers based on enhanced permeability and retention (EPR) effects is confused by heterogeneity of the microenvironment in tumor [11].

Nanoparticles (NPs) such as gold (AuNPs) have been well investigated and widely used in carrying drugs or biomolecules for cancer therapies owing to easiness of fabrication, tunable sizes, biocompatibility, distinct physicochemical characteristics, and easy functionalization with biomolecules [12]. In spite of the versatile advantages of AuNPs, nanoparticles-based therapies still have several limitations, such as permeability, vascular barriers, and drug retention. Consequently, several methods, such as surface modification with biopolymers or penetrated molecules on nanoparticles, have been explored in recent years to facilitate stability, intracellular targeting, and cellular uptake capacity so as to improve the efficiency of drug utilization [13]. For instance, collagen is the major structural protein and plentiful in connective tissues, such as bones, skin, and blood vessels [14]. Traditionally, Au is used for treating rheumatism, and the effects may derive from interacting with collagen. With the advantage of strength and stability through self-aggregation, as well as crosslinking, collagen has garnered interest as a potential nano-biomaterial. Advanced biotechnology presently supports this material to be used in drug delivery. Research recently revealed that collagen molecule tends to have rapid association with gold nanoparticles, and undergo unfolding of the helical architecture by adsorption on AuNPs [15]. For the biocompatibility and biodegradability, collagen-AuNPs composite (Col-Au) may serve as a promising platform for delivering bioactive molecules [16].

The endocytic pathways have been classified into various types: entosis, phagocytosis, pinocytosis, and circular dorsal ruffles [17]. Based on the researches in cellular uptake mechanisms of nanoparticles, most are associated with phagocytosis and pinocytosis. Previous studies indicate that phagocytosis can be upregulated while the cells undergoing microbial invasion, debris, or the size of nanoparticle is over 500 nm. Moreover, the detailed classifications of pinocytosis pathway are presented as clathrin and caveolae-independent endocytosis, micropinocytosis, clathrin-mediated endocytosis, and caveolae-mediated endocytosis [18]. The internalization of AuNPs in cells has attracted interests for studying on the endocytic pathways to improve the delivery efficiency of nanoparticle-mediated drugs and biomolecules. Nanomedicine delivery systems have been extensively investigated in past decades. The bilayer phospholipid systems, i.e., liposomes, were firstly investigated in 1965 and then suggested as an efficient drug delivery system [19]. There are various technical advancements in liposome delivery system, such as remote drug loading, long circulating (PEGylated) liposomes, liposomes comprising nucleic acid polymers, and liposome containing different drugs. The above advancements facilitate the parenteral drug administration in delivery of anti-cancer, antibiotic drugs, and also gene medicines. For example, the liposomal Doxil^®^ exhibited lower systemic toxicity, and improved pharmacokinetic properties in breast cancer treatments [20].

Cancer therapy using natural compounds and nutraceuticals for clinical treatments have gained much attention due to lower cytotoxicity and decreasing side effects of conventional chemotherapy [21]. Natural products have been considered to be an efficient anti-neoplastic agent for anticancer pharmaceuticals [22]. There are a variety of bioactive plant-derived compounds that receive attentions due to potential abilities in preventing, suppressing, and reversing the progression of cancer diseases [23]. Berberine (denoted as BB), an isoquinoline alkaloid compound, is extensively discovered in medicinal plants and applied for traditional Chinese medicine [24]. BB is verified to stimulate apoptosis and inhibit cell proliferation in breast, lung, and colon cancer cell lines [25,26]. BB can also suppress tumor growth by reducing proliferation of abnormal cell, arresting cell cycle progression, and stimulating apoptosis in breast cancer and osteosarcoma cells [27,28]. A recent research proved BB to be an autophagy suppressor that could inhibit the formation of autophagosome in MCF-7/ADR breast cancer cell lines. BB inhibited autophagy through regulating PTEN to affect PI3K/Akt/mTOR pathway and also inhibited autophagy-associated protein LC3II to accumulate, which lead to p62 cellular accumulation, reduction in cell proliferation, and reversal of doxorubicin resistance [29]. Another study demonstrated that BB combined with cisplatin suppressed the breast cancer cell proliferation through causing DNA double-strand breaks and caspase-3-dependent apoptosis, indicating BB could significantly influence cell proliferation, cell cycle progress, and apoptosis in both T47D and MCF7 human breast cancer cell lines. Indeed, BB and doxorubicin alone or in combination remarkably stimulated apoptosis in both cell lines. BB also enhanced the anti-tumor capacity of tamoxifen in drug-sensitive MCF-7 and drug-resistant MCF-7/TAM cells. Studies elucidated the combinational treatment was efficient in inducing G1 phase arrest and stimulating apoptosis owing to upregulating the expression of p21 and downregulating of Bcl-2 protein [30]. In another work, BB suppressed the migration of MCF-7 breast cancer cells via reducing the mRNA levels of chemokine receptors, supporting BB to become a potential biomolecule to target chemokine receptor genes in human breast cancer metastasis [31].

Literature confirms BB as a superior chemotherapeutic drug for breast cancer treatments. In addition, biocompatible AuNP nanocarriers have been fabricated through conjugating AuNPs with collagen (Au-Col) and demonstrated the enhancement effects on MSCs proliferation, migration and differentiation abilities [32]. In the current research, the AuNP-collagen-berberine (Au-Col-BB) nanocarriers were prepared. These hybrid nanocarriers were subjected to evaluation of physicochemical properties, examination of cytotoxicity, investigation of cellular uptake mechanisms, and xenograft mice models in non-transformed BAEC and Her-2 breast cancer cell lines to verify their potential in breast cancer therapies.

## 2. Materials and Methods

### 2.1. Material Preparation of Gold-Collagen-Berberine (Au-Col-BB)

The physical gold nanoparticles were purchased from Gold NanoTech, Inc. (GNT, Taipei, Taiwan). The GNT Gold is 99.99% pure, manufactured by physical vapor deposition (PVD) processing, which is different from chemical synthesis used by commercially available nanogold products. GNT Gold contains no other heavy metals or toxic compounds.

The synthesis of gold nanoparticles in cooperation with berberine extracts is described below. Au-Col-BB conjugates were synthesized following a modified procedure. AuNP-collagen-berberine nanocarrier (Au-Col-BB) were firstly prepared by fully mixed 100 μL of AuNP (Gold Nanotech Inc., Taipei, Taiwan) with 100 μL of collagen (0.5 mg/mL, (BD Bioscience, Canton, MA, USA)) for 30 min at RT to obtained the Au-Col nanocomposites (200 μL, 20 ppm). Next, Au-Col nanocomposites were interacted with berberine (0.5, 1, 5, and 10 μg/mL) in 3:2 vol ratio to obtained the AuNP-Col-BB (at 4 °C for 2 h). For FITC conjugation, Au-Col-BB were combined with 0.5 mg/mL FITC (Sigma-Aldrich, Burlington, MA, USA) at 4 °C for 8 h in a 50:1 vol ratio. Au-Col-BB-FITC were cautiously washed two steps with DI water and stored at 4 °C in a dark. Berberine (99.8% purity) was obtained from Sigma-Aldrich Co. Au-Col-BB were prepared by our patented method according to our previous study [15].

### 2.2. Material Characterization

The physical and chemical properties of the physical gold and fabricated gold nanoparticles, namely Au, Au-Col, and Au-Col-BB, were further characterized by using methodology in our previous research [32]. The UV-Vis absorption spectrum was obtained through Helios Zeta spectrophotometer (Thermo Fisher, Pittsfield, MA, USA). The wavelength ranges from 190 to 1100 nm, and the typical absorption peak for Au is 520 nm. The measured data were analyzed by using Origin Pro 8 (Originlab Corporation, Northampton, MA, USA) software. Afterwards, Fourier transform IR spectrometer (Shimadzu IRPretige-21, Japan) was used to obtained the detailed spectra information about functional groups at the frequency range 400 to 4000 cm^−1^. For FTIR measurement, the mixture of 1% (*w/w*) sample and 100 mg of KBr powder (Sigma-Aldrich, Burlington, MA, USA) was pressed into a sheer slice. An average of 32 scans for each measurement was obtained with 2 cm^−1^ resolution so as to improve the signal-to-noise percentage. Furthermore, dynamic light scattering (DLS) analysis was proceeded with Malvern Zetasizer Nano ZS and manipulating at a light source (532 nm) with 90° fixed scatter angle. The measurements were processed after adding 1 mL of the colloidal sample in a cuvette with 1 cm optical path. To investigate the size distribution, the cumulant method was applied to analyze the intensity distribution values. To determine the size, zeta potential, size distribution, and polydispersity index (PDI) of NPs, the solutions were moved into transparent cuvettes. Pure NPs and BB-loaded NPs were investigated through a Zetasizer Nano ZS (Malvern Panalytical Ltd., Malvern, UK) instrument coupled with a 633 nm He–Ne laser at 25 °C for following investigations.

### 2.3. Cell Culture

The A549 lung adenocarcinoma cell line and Colo-205 Colorectal cancer cell line were obtained from American Type Cell Culture (ATCC). The cells were cultured in the DMEM or RPMI 1640 medium supplemented with 10% FBS (GIBCO), 1% antibiotic solution (penicillin 5000 U/mL and streptomycin 5000 μg/mL) and 1% L-glutamine (200 mM). Her-2 cells, a human breast adenocarcinoma cell line, were stored in DMEM culturing medium containing 10% heat-inactivated fetal calf serum and 1% P/S. Bovine aortic endothelial cells (BAEC) were cultured with DMEM containing 5.5 mM D-glucose (low glucose) and 4 mM L-glutamine supplemented with 10% FBS, 2.5 mg/L ECGS, 1% MEM non-essential amino acids, and pen-strep-neomycin in incubator (37 °C, 20.7% O_2_, 5.0% CO_2_).

### 2.4. Examination of Cell Viability

The MTT (3-(4, 5-cimethylthiazol-2-yl) 2, 5-diphenyltetrazolium bromide) (Sigma-Aldrich, Burlington, MA, USA) was purchased to further investigate cytotoxicity following with the manufacturer’s instructions. Briefly, A549, Colo-205, BAEC and Her-2 cells at a density of 8 × 10^3^ per well were seeded in 96-well plate and treated with various concentrations of BB alone (1 μg/mL) or Au-Col-BB (0.5, 1, 5, and 10 μg/mL). After the above treatment, 20 μL MTT (0.5 mg/mL) was added and incubated for 4 h (37 °C). Next, removed cell supernatants. DMSO was added to dissolve the crystals, and the absorbance setting up at 570 nm was detected through TECAN ULTRA microplate reader. The relative cell viability (%) was analyzed through the absorbance of treated samples compared to the untreated control group (represented as 100%). The IC_50_ values represented as the concentration that cause 50% of cell growth inhibition evaluated by using 95% confidence intervals (95% CI) from non-linear regression after normalize the results compared to untreated control group. Each experiment was triplicated. The results of the MTT assay were used to calculate the 50% inhibitory concentration (IC_50_).

Cells at a density 80–95% were labeled through using free 2 μM Calcein AM (Sigma-Aldrich, Burlington, MA, USA) solution. After incubating for 30 min, cells were washed with PBS and then added new medium. Cells were incubated with different concentrations of Calcein AM (2–8 μM) for 30 min at 37 °C following manufacturer’s instructions (*n* = 8 in each concentration). Next, the cells were washed three times by fresh culture media. The images of cells were captured via using a fluorescence microscope (Olympus Corp., Shinjuku-ku, Tokyo, Japan) and the fluorescence intensity was measured and analyzed through Image J 5.0 software.

### 2.5. Cellular Uptake Assay

To investigate cellular uptake of the nanoparticles, cells (1 × 10^5^) were cultured with Au-Col-BB-FITC (0.5 mg/mL) for 2 h then using PBS to wash away excess nanoparticles. The cells were incubated at 37 °C for various times (30 min, 2 and 24 h). Afterwards, the cells were washed by using phosphate-buffered saline (PBS) and fixed with 4% paraformaldehyde (PFA) for 15 min, 0.5% Triton X-100 (Sigma-Aldrich, Burlington, MA, USA) for permeability for 10 min, and F-actin staining (6 μM Rhodamine phalloidin, Sigma-Aldrich). After further incubation, the nucleus was stained for 10 min with 4, 6-diamidino-2-phenylindole (DAPI) nuclear staining (1 mg/mL, Invitrogen, Sigma-Aldrich, Burlington, MA, USA) and washed 2 times with PBS. The fluorescence was visualized by Zeiss Axio Imager A1 fluorescence microscope, and were quantified by using Image J 5.0 software. Furthermore, the cellular uptake ability was investigated by detecting fluorescein-positive cells through flow cytometer and quantified via fluorescence-activated cell sorting (FACS) software (BD Biosciences, Canton, MA, USA).

To evaluate the endocytosis pathways of cellular internalization of Au-Col-BB, the cells were cultured with different specific inhibitors of cell energy-dependent endocytosis. To investigate the cellular uptake mechanisms, cells at a density of 1 × 10^5^ were cultured with different endocytosis inhibitors, such as Cytochalasin D (Cyto-D, 13.1 μM), Chlorpromazine (CPZ, 7.5 μM), Bafilomycin (Baf, 382.3 nM) and 2-Mercaptoethanol (β-ME, 14 mM) for 1 h at 50% inhibitory concentration (IC_50_). All inhibitors were prepared in 10% DMSO, vortex for 10 s, and centrifuged (10 s; 14,000× *g*) before dilution in culture medium, and cultured with Au-Col-BB-FITC (0.5 mg/mL) for 2 h. After using PBS to remove the residual nanoparticles and incubating in medium for various time (30 min, 2 and 24 h) at 37 °C, cells were collected through using a centrifuge (7000 rpm, 4 °C, 5 min), and resuspended in PBS. The fluorescence intensity of 1 × 10^4^ cells was quantified by FACS method (BD Biosciences, Canton, MA, USA). Those fluorescein-positive cells were quantified by fluorescence activated cell sorting (FACS) Calibur flow cytometer (BD Biosciences, Canton, MA, USA), and data were analyzed through Flow J software 7.6.1. All experiments were represented in triplicate.

### 2.6. LysoTracker Assay

A fluorescence microscopy can be applied to observed the intracellular delivery of polymer micelles, such as endosomal/lysosomal escape. In current research, a fluorescence microscopy was used to investigate the cytoplasmic distribution of Au-Col-BB in cells. The localization of Au-Col-BB-FITC (green fluorescence) was observed through labeling cells with LysoTracker fluorescent probes (red color). Cells (1 × 10^5^/well) were cultured in a 24 well plate (37 °C, pH 6.5) and incubated with Au-Col-FITC and Au-Col-BB-FITC at concentration of 1 μg/mL for various times (30 min, 2 and 24 h). Afterwards, cells were washed thrice by 4 °C PBS and stained with 50 nM LysoTracker Red (Invitrogen) at 37 °C for 30 min. Hereafter, the cells were washed thrice with PBS and observed under fluorescence microscopy. All experiments were performed in triplicate.

### 2.7. Cell Cycle Analysis

Cells at the density of 2 × 10^5^ were seeded in 6-well plates and cultivated for attachment for 48 h. Thereafter, previous media was removed and the new culture media containing BB alone (1 μg/mL), Au-Col (0.5 mg/mL) and Au-Col-BB (1 μg/mL) was added for 48 h incubation. After treatment, the media was removed and PBS solution was applied for twice washing steps prior to trypsinization. The cells were collected through a centrifuge (1500 rpm, 5 min), next, suspending in 70% ethanol (1 mL, −20 °C) for 20 min. Afterwards, the cells were resuspended in 500 μL of PBS, incubated with RNase A (40 μg/mL) and stained with PI (10 μg/mL). Before processing analysis, the stained cells were incubated in dark for 30 min at RT. BD LSRFortessa™ Cell Analyzer (BD Biosciences, Canton, MA, USA) was applied to determine the cell cycle progress for 10,000 cells in each sample. The results analyzed through BD FACSDiva™ software was presented by percentage of cells compared to the populations of the untreated control. All experiments were processed in triplicate.

### 2.8. Annexin V-PI Staining Assay

Apoptosis cell induced by Au-Col-BB was detected with fluorescence staining using an Annexin V-FITC/propidium iodide (PI) apoptosis detection kit (BD Pharmingen). The cells were fixed with 3.7% formaldehyde and permeabilized with 0.1% Triton X-100 in PBS. Quantification of apoptotic cells was examined by the adherent cells collected by centrifugation. The cells were stained with Annexin-V and PI (both 5 μL) and incubated in the dark (15 min at RT). For nuclear staining, cells were incubated with DAPI (1 µg/mL) for 30 min at 37 °C. Afterwards, the stained cells were washed with PBS and observed by using fluorescence microscope and were analyzed through a flow cytometry. For detailed, Annexin V^+^/PI^−^ cells were considered to be in the early stage of apoptosis, Annexin V^+^/PI^+^ cells were in the late stage of apoptosis, Annexin V^−^/PI^+^ cells were defined as necrotic cells, and Annexin V^−^/PI^−^ cells were normal cells. Dead cells and apoptosis/necrosis cells were detected by a BD FACS Calibur flow cytometer (BD Biosciences, Canton, MA, USA). The cells in early stages of apoptosis were Annexin V-positive and PI-negative, and the cells in the late stages of apoptosis were both Annexin V and PI-positive. The acquisition of results and analysis were carried out with a Becton-Dickinson FACS Calibur flow cytometer through Flow J software. All experiments were performed in triplicate.

### 2.9. Gelatin Zymography Analysis

The collected samples were separated by a 10% zymogram protein gel containing 0.1% gelatin. The gelatin degrading proteolytic activities of the MMPs on the gel were detected described in previous protocols [33]. The samples were centrifuged, the supernatants were next collected and stored at −80 °C for following MMP-2 and MMP-9 determination. Bicinchoninic acid protein assay reagents (Pierce, WA, USA) was applied to determine the protein concentrations. The enzymatic activities of matrix metalloproteases, MMP-2 and MMP-9, were investigated by gelatin zymography in gels. In short, 25 µg of total protein was separated in SDS-PAGE containing 1 mg/mL gelatin. Next, the gels were treated with renaturing buffer (2.5% Triton X-100) for 30 min at RT to remove SDS. Zymograms were subsequently incubated overnight at 37 °C in developing buffer (50 mM Tris Base, pH 7.5, containing 200 mM NaCl, 5 mM CaCl_2_.2H_2_O, and 0.02% Brilj-35). The gels were stained by using 0.5% Coomassie Blue and destained by 10% acetic acid and 40% ethanol. The densitometric analysis of lytic bands through MMP gelatinase activity was analyzed by Image J 5.0 software (Media Cybernetics Inc., Rockville, MD, USA).

### 2.10. Assessment of Cell Migration Ability

The cell migration ability was investigated through an Oris Cell Migration Assay reagent kit (Platypus Technologies, Madison, WI, USA). Firstly, the cells were added into an Tri-Coated plate, as following report [33]. The stoppers were seeded in each well with 100 µL at the cell density of 8 × 10^3^ cells/mL to reach confluent after 48 h. Next, we removed all the stoppers but retained one for pre-migration. The plate was incubated at 37 °C for investigation of pre-migration (t = 0 h) and post-migration (t = 24 and 48 h). Next, cells were stained with 2 µM Calcein AM (Sigma) with serum free medium. The images were captured using Zeiss Axio Imager A1 fluorescence microscope. The distance of cell movement into the detection area were semi-quantified based on fluorescence intensity analyzed through Image J 5.0 software.

### 2.11. Western Blot Assay of Apoptotic Protein Expression

For Western blotting assay, the cells were incubated with different materials for 48 h. Cell pellets were collected and incubated in RIPA lysis buffer containing protease inhibitor. The total proteins were extracted by lysis buffer (50 mM Tris, pH 7.4, 1 mM EDTA, 1 mM Phenylmethylsulfonyl fluoride, 25 mg/mL leupeptin, 0.1 mg/mL aprotinin, 1 mM dithiothreitol, 1 mM NaF, and 1% NP-40), and the protein concentration was measured through BCA Protein Assay Reagent Kit. In accordance with the protocol (Bio-Rad Laboratories Inc., USA), the protein samples (25–30 μg) were electrophoresed on a 10% sodium dodecyl sulfate polyacrylamide separation gel and transferred onto polyvinylidene fluoride membranes (Immobilon P; EMD Millipore). After blocking in TBST with 5% milk powder for 1 h, the membranes were probed with primary antibodies, 1: 1000 dilution of Bcl-2, Bax, Cyclin D1, p21 (1:1000 dilution, Santa Cruz, Dallas, TX, USA), β-actin (1:5000 dilution, Santa Cruz, Dallas, TX, USA) at 4 °C overnight, and with the fluorescent secondary antibodies at 25 °C for 1 h. The membranes were washed thrice with TBST between each incubation. The immunoblots were washed thrice with TBS-T and incubated with HRP-conjugated goat anti-rabbit or anti-mice IgG (1:2000 dilution) (Zhongshan Goldenbridge Biotechnology, Xuanwu District, Nanjing, China) at RT for 1 h. Immunoblots were measured by using an ECL kit (Beyotime Institute of Biotechnology, Shanghai, China) and visualized after exposure to X-ray film. The protein expression levels were measured through using Image Quant 5.0 software (Molecular Dynamics, Caesarea, Israel). Each protein band density was normalized to β-actin.

### 2.12. Tumor Xenograft Mouse Model

This study was cautiously processed with the approvals in the Guide for the Care and Use of Institutional Animals of China Medical University and the Care and Use of Laboratory Animals of the National Institutes of Health. Animal housing, care, and application of experimental procedure were conducted based on a protocol approved by the Institutional Animal Care and Use Committee of the China Medical University. The 20–25 g male BALB/c nude mice (approximately 2 months) were obtained from the National Laboratory Animal Center (Taiwan). Her-2 cells (2 × 10^6^) in 50 μL matrigel were inoculated subcutaneously into the flanks of each mouse. Tumor growth was investigated through Vernier caliper and the tumor volume (V) was calculated based on the formula “V (mm^3^) = (D1^2^ × D2)/2” (D1 and D2 represented as the shortest and longest tumor diameter). After the tumor was approximately 25 ± 1 mm^3^ (day zero), the following treatments were processed. All the tumor-bearing mice were randomly assigned into six groups: Au-Col, Au-Col-BB (5, 10, and 20 mg), control group (just treated with PBS), and BB only (20 mg). Each group included 6 mice (*n* = 6). Then, those formulas were given to mice on the 8th day, 11th day, and 14th day via the tail vein (intravenous) injection (50 μL). During the treatment period, the body weight and tumor size were measured every other day. After 30 days of treatment, mice were sacrificed through decapitation and tumors are removed and measured. In the survival time experiment, as in the previous experiment method, five mice in each group (*n* = 5) were evaluated for survival time until the 70th day. The median survival of mice in all treatment group was calculated using Kaplan–Meier statistics and log-rank test.

### 2.13. Statistics Analysis

In current research, the experiments were independently repeated triplicates for uncertainty avoidance. Data from each sample (*n* = 3–6) were collected and expressed as mean ± standard deviation (SD). Student’s *t*-test and the single-factor analysis of variance (ANOVA) method were applied to measure the difference between groups. Bonferroni was chosen as post hoc analysis for ANOVA method. The *p* value less than 0.05 was considered as statistically significant (* *p* < 0.05, ** *p* < 0.01, *** *p* < 0.001).

## 3. Results

### 3.1. Material Characterization of the AuNP-Collagen-Berberine (Au-Col-BB) Nanocarriers

The brief concept elucidates the procedures for manufacturing gold nanoparticles conjugated with collagen and berberine (Figure 1A). The energy absorption shift for decorated particles were investigated by UV-Vis spectrophotometer. Transmission electron microscope (TEM) was further applied to observe material surface of Au (a), Au-Col (b), and (c) Au-Col-BB, demonstrating the spherical shape of nanoparticles (Figure 1B). The DLS assay revealed the size distribution intensity of AuNPs, Au-Col, Au-Col-BB, and Au-Col-BB-FITC (Figure 1C). Next, the surface potential of different nanoparticles was also evaluated by DLS analyzer. Zeta-potential was −8.9 for gold nanoparticles (Au alone), which became positive potential of 5.92 mV with collagen adsorption, and then returned to negative potential of −2.92 mV after cross-linking with BB. Further, zeta-deviation of pure Au was 3.88 mV, decreased to 3.72 mV after Au combined with collagen, then the lowest in Au-Col-BB, 3.76 mV. The polydispersity index (PDI) of each nanoparticle was also investigated, as the value of pure Au was 0.43, the Au-Col nanocomposite was 0.48, and the Au-Col-BB nanocarrier was the highest, 0.65 (Figure 1C). The size of derived gold nanoparticles was measured by DLS analyzer. The diameter was approximately 24.5 nm for Au alone, with collagen adsorption, the size increased to 196 nm (*p* < 0.01), and further conjugated with BB that became 227 nm. Obviously, the size remarkably increased after incorporating with BB (*p* < 0.01) (Figure 1D).

The UV-Vis absorption peak at 520 nm was observed for the gold-containing nanoparticles, Au, Au-Col, and Au-Col-BB, indicating the presence of Au in each sample (Figure 1E). The FTIR spectrum of various materials were represented in Figure 1F. The peak at 1650.7 cm^−1^ was corresponded to the amide I vibration (C=O) and the amide II (CH_2_, CH_3_), amide III (C–N, N–H) adsorption peaks were located at 1544.34 cm^−1^ and 1243.47 cm^−1^ which was typical of collagen [34,35]. After collagen conjugated with Au nanoparticles, the position of amide I peak in pure collagen was shifted to 1654.63 cm^−1^. The pure berberine compound was also be investigated, the results showed the C-H stretching was found at 2924.74 cm^−1^, while C=C and C=N stretching was observed at 1638.40 cm^−1^. The deformation in C-H was found from 1354 to 1383 cm^−1^ and C–O stretching was found at 1096.40 cm^−1^ [36]. Next, Au-Col was cross-linking with berberine, there was a shift in the peak position of amide I from 1654 cm^−1^ to 1655.40 cm^−1^, amide II from 1544.32 cm^−1^ to 1545.71 cm^−1^. The above result indicated that Au-Col nanoparticle may interact with berberine.

### 3.2. Assessment of Au-Col-BB on Cell Viability and Cytotoxicity on A549, Colo-205, and Her-2 Cell Lines

By MTT assay, the effects of Au-Col-BB on cytotoxicity were investigated with A549, Colo-205 and Her-2 cancer cell lines. Cells were cultivated in 96-well dishes. After attachment, these cell lines were treated with Au-Col-BB (1, 5, and 10 μg/mL) and BB (1 μg/mL) for 48 h. Figure 2A demonstrated Au-Col-BB and BB alone induced significant cytotoxicity on three cancer cell lines till 48 h, particularly for Au-Col-BB (10 μg/mL) and BB (1 μg/mL). The amount of A549 decreased to 0.62 fold (*p* < 0.01) in Au-Col-BB (1 μg/mL), 0.37 fold (*p* < 0.01) in Au-Col-BB (5 μg/mL), 0.2 fold (*p* < 0.01) in Au-Col-BB (10 μg/mL), and 0.23 (*p* < 0.01) fold in BB (1 μg/mL); the amount of Colo-205 decreased to 0.53 fold (*p* < 0.001) in Au-Col-BB (1 μg/mL), 0.29 fold (*p* < 0.001) in Au-Col-BB (5 μg/mL), 0.2 fold (*p* < 0.001) in Au-Col-BB (10 μg/mL), and 0.23 fold (*p* < 0.001) in BB (1 μg/mL); in Her-2 cell line, the amount also significantly decreased to 0.53 fold (*p* < 0.01) in Au-Col-BB (1 μg/mL), 0.34 fold (*p* < 0.001) in Au-Col-BB (5 μg/mL), 0.22 fold (*p* < 0.001) in Au-Col-BB (10 μg/mL), and 0.23 fold (*p* < 0.001) in BB (1 μg/mL), respectively, compared to controls. In order to evaluate the effects of Au-Col-BB on cell viability, Calcein-AM assay was applied to perform the experiment. The fluorescence intensity results (Figure 2B) also revealed that the cell viability was remarkably inhibited by Au-Col-BB (1 μg/mL) and BB (1 μg/mL): A549 (0.58 and 0.38 fold, *p* < 0.001), (Colo-205 (0.47, *p* < 0.01) and 0.27 fold (*p* < 0.001)), and Her-2 (0.58 and 0.37 fold, *p* < 0.001). Furthermore, the Annexin V-PI staining assay (Figure 2C) was applied to observe the amount of apoptotic (Annexin-V^+^ fluorescent intensity) and dead cells (PI^+^ fluorescent intensity) after cultured with BB (1 μg/mL) and Au-Col-BB (1 μg/mL). The quantification results were showed in Figure 2D. In A549 culturing with BB (1 μg/mL) and Au-Col-BB (1 μg/mL), the apoptotic cells (282.7 and 300.2) were both more than dead cells (127.2 and 121.7). In contrast, the apoptotic and dead cells (379 and 315.2) of Colo-205 treating with Au-Col-BB (1 μg/mL) were more than BB (1 μg/mL) group (149.2 and 157.2). In addition, in Her-2 cell line, the amounts of apoptotic cells treating with BB (1 μg/mL) were 210 and 204, the dead cells were calculated as 223 and 189.2, respectively. The above results indicated Au-Col-BB could significantly inhibit cell proliferation and induce cell apoptosis in A549 lung cancer cell and Colo-205 human colon cancer cell lines. Further, Her-2 breast cancer cells were treated with Au-Col-BB and BB to evaluate the cytotoxicity. We found that Au-Col-BB also had the potent for inhibiting Her-2 cell growth and increasing the amount of apoptotic and death cells, verifying the efficiency of cytotoxicity in breast cancer cells.

### 3.3. Investigation of Biocompatibility in BAEC and Her-2 Cell Line Treating with Various Materials

To investigate the biocompatibility of Au-Col-BB between normal and cancer cells, non-transformed BAEC were chosen to compare with Her-2 breast cancer cells. Both BAEC and Her-2 cell line were treated with BB (1 μg/mL), and Au-Col-BB (0.5, 1, 5, and 10 μg/mL) for various time points (24, 48 and 72 h) to analysis cell viability and cell cycle influence. Figure 3A,B demonstrated that BB and Au-Col-BB could induced a significant cytotoxicity on Her-2 cell line, particularly in Au-Col-BB (10 μg/mL) group at 24, 48 and 72 h.

For non-transformed BAEC, the sub-G1 population was slightly increased after treating with BB (1 μg/mL) and Au-Col-BB (0.5, 1, 5, and 10 μg/mL) compared to the control (Figure 3C). To view cell cycle influence, Her-2 cancer cells treated with Au-Col-BB especially for 10 μg/mL demonstrated the greatest sub-G1 cell populations (*p* < 0.01) compared to the control (Figure 3D). Further, Appendix A demonstrated the cell viability of both BAEC and Her-2 cell line culturing with various concentrations of BB (0.5, 1, 5, and 10 μg/mL), the results indicated treating with BB 10 μg/mL had the lowest cell viability compared to other groups. The population of apoptotic cells for BAEC and Her-2 cells treating with various materials was evaluated by Annexin V-PI double staining assay. The images of flow cytometry were displayed as Figure 4A (BAEC) and Figure 4B (Her-2). In Figure 4D, the population of apoptotic (Annexin-V positive) Her-2 cells was 80.6%, which was the highest induced by Au-Col-BB (10 μg/mL). Additionally, the population of viable Her-2 cells was the lowest (18.2%) after treating with Au-Col-BB (10 μg/mL). On the contrary, the quantitative results in Figure 4C indicated that after BAEC treating with Au-Col-BB (10 μg/mL), the population of apoptotic cells (Annexin-V positive) was 0.78%, and the population of viable cells was 95.63%. The above evidence showed Au-Col-BB (10 μg/mL) could efficiently induce cell apoptosis in Her-2 cells, supporting that Au-Col-BB had specific cytotoxicity effect.

### 3.4. Investigation of Cell Uptake Efficiency between Her-2 and BAEC Cell Lines

The uptake efficiency at 2 h was also evaluated by using non-transformed BAEC (Figure 5A) and Her-2 breast cancer (Figure 5D) cell lines which was treated with FITC conjugated Au-Col and Au-Col-BB. Next, we observed the cells by using fluorescent microscope. The fluorescent images demonstrated the nanoparticles were presented inside cells at 2 h. In addition, immunofluorescence method (IF) and fluorescence-activated cell sorting (FACS) were also applied to quantify the uptake efficiency in both cell lines. According to the results of IF method for BAEC, the uptake amount of Au-Col and Au-Col-BB was increased to ~4.01 and ~4.96 fold at 2 h, and achieved to ~7.11 and ~9.17 fold at 24 h, compared to the control group. In addition, the uptake amount of Au-Col and Au-Col-BB determined by FACS demonstrated similar trend, which was ~5.43 and ~8.45 fold at 2 h, and increased to ~6.95 and ~12.3 fold at 24 h, respectively (Figure 5B,C). Further, the uptake amount in Her-2 cell line was also measured. In IF, the uptake amount for Au-Col was ~4.2 fold at 2 h, ~5.72 fold at 24 h; and for Au-Col-BB was ~4.67 fold at 2 h and ~6.15 fold at 24 h (Figure 5E). However, the FACS results indicated the uptake amount of Au-Col and Au-Col-BB was both significantly higher at each time point compared to the control. For Au-Col was ~2.1 and ~2.2 fold at 2 and 24 h. For Au-Col-BB group was ~2 fold at 2 h, and ~2.1 fold at 24 h, respectively (Figure 5F). The images of uptake efficiency at 30 min and 24 h was also evaluated by using non-transformed BAEC Appendix A and Her-2 breast cancer cells Appendix A which was treated with FITC conjugated Au-Col and Au-Col-BB. The results elucidated after the nanocarrier Au-Col-BB could be significantly uptake by both BAEC and Her-2 cells compared to Au-Col.

### 3.5. Assessment of Potential Cellular Transportation between Her-2 and BAEC Cell Lines

Lysosome is the major organelle involving with foreign body entry cells. To investigate lysosomal effects on Au-Col-FITC and Au-Col-BB-FITC, lysosomal tracker was applied to target lysosome of BAEC (Figure 6A–C) and Her-2 (Figure 6D–F) cell lines. The green, red, and blue fluorescence represented as gold nanocarriers, lysosomes, and cell nuclei at 2 h were shown in Figure 6A,D. Based on the IF method in BAEC cell line, the FITC intensity of Au-Col and Au-Col-BB were increased to ~4.4 and ~4.9 fold at 2 h, and further increased to ~5.2 and ~5.2 fold at 24 h (Figure 6B). The semi-quantitative results from FACS method also indicated the increased intensity of Au-Col and Au-Col-BB at both 2 h (~4.48 fold and ~4.92 fold) and 24 h (~5.2 fold and ~5.28 fold) (Figure 6C). Furthermore, the FITC intensity for Her-2 cell line was also investigated. The uptake amounts in Her-2 at 30 min quantified by IF method was ~4.67 fold (Au-Col) and ~6.41 fold (Au-Col-BB). However, the uptake amount of Au-Col-BB was decreased to ~4.84 fold and Au-Col was ~5.04 fold at 2 h. Further, at 24 h, the intensity of Au-Col increased to ~6.76 fold, and the intensity of Au-Col-BB was slightly decreased to ~4.69 fold (Figure 6E). In contrast, the FACS method indicated different results that the intensity of Au-Col-BB in Her-2 cells (~4.98 fold) were greater than Au-Col (~4.65 fold) at 2 h. The similar trend was also observed at 24 h (Au-Col-BB: ~5.25 fold, Au-Col: ~4.98 fold) (Figure 6F). Additionally, the images of FITC fluorescence intensity were also demonstrated at 30 min and 24 h by using non-transformed BAEC Appendix A and Her-2 breast cancer cell line Appendix A after treating with FITC conjugated Au-Col and Au-Col-BB. Thus, based on the results of Lysotracker assay, we observed that Au-Col-BB would not degrade in lysosome after the autophagy and was seen to be stable, supporting the high drug delivery capacity of Au-Col carrying BB.

Above all, endocytosis is the pathway that cell absorbs foreign bodies. The endosome undergoes transporting and metabolizing with lysosome when foreign bodies enter cells. In present study, 4 different lysosomal inhibitors: Cyto-D, CPZ, β-ME, and Baf, were using to further investigate the mechanisms of lysosomal effects involved in metabolic process. In addition, BAEC cells showed the less effects after culturing with lysosomal inhibitors at 2 h (Figure 7A), but the quantification results measured by IF and FACS methods indicated that the uptake ability was also significantly inhibited by CPZ and Baf. The IF results indicated (CPZ: 30 min: (~0.71), 2 h: (~0.87), and 24 h: (~0.87) fold, Baf: 30 min: (~0.91), 2 h: (~0.97), and 24 h: (~0.89) fold) (Figure 7B) while the FACS method demonstrated [CPZ: 30 min: (~0.68), 2 h: (~0.82), and 24 h: (~0.84) fold, Baf: 30 min: (~0.87), 2 h: (~0.9), and 24 h: (~0.85) fold], respectively (Figure 7C). After we treated the Her-2 cancer cell line with each lysosomal inhibitor, the IF images demonstrated that Au-Col-BB-FITC and lysosome colocalization were significantly faded at 2 h particularly in CPZ and Baf treatment (Figure 7D). Distinctly, the quantification of IF (Figure 7E) indicated that CPZ and Baf exhibited the strongest inhibition effects to interfere lysosomal metabolic effects on Au-Col-BB nanoparticles at various times compared to control (CPZ: 30 min: (~0.46), 2 h: (~0.35), and 24 h: (~0.27) fold, Baf: 30 min: (~0.27), 2 h: (~0.27), and 24 h: (~0.4) fold). In addition, the FACS method also figured out similar situation (CPZ: 30 min: (~0.38), 2 h: (~0.3), and 24 h: (~0.34) fold, Baf: 30 min: (~0.35), 2 h: (~0.32), and 24 h: (~0.28) fold) (Figure 7F). The fluorescence images were also demonstrated by using non-transformed BAEC Appendix A and Her-2 breast cancer cells Appendix A at 30 min and 24 h which was treated with FITC conjugated Au-Col and Au-Col-BB. Appendix A elucidated the cell viability of Her-2 cell after treated with 4 different lysosomal inhibitors: Cyto-D, CPZ, β-ME, and Baf, to confirm 50% inhibitory concentration (IC_50_). The results showed that value of IC_50_ between each lysosomal inhibitor: Cyto-D (13.1 μM), CPZ (7.5 μM), β-ME (14 μM), and Baf (382.3 nM). According to the above results, we investigated the mechanism of endocytotic route in both BAEC and Her-2 cell line and found that the uptake of Au-Col-BB nanocarrier could be significantly inhibited by CPZ and Baf lysosomal inhibitors due to the size of Au-Col-BB nanocarrier (227 nm), while CPZ inhibited clathrin-mediated endocytosis and Baf interfered cell autophagy.

### 3.6. Assessment of Migratory Inhibition Effects

To investigate cell migration influence after Au-Col-BB treatment, the Oris™ cell migration and MMPs activity assay was applied to investigate migration capacity. The zymography images of BAEC and Her-2 were shown as Figure 8A,B. Based on the quantitative results in Figure 8C for BAEC, there was no significant difference in both MMP-2/9 compared to the control at 48 h. In Figure 8D, for the Her-2 cell line, the results indicated that the expression of MMP-9 were the lowest in Au-Col-BB (10 μg/mL) at 48 h. Additionally, the expression of MMP-2 was significantly decreased in BB (1 μg/mL), Au-Col-BB (5, 10 μg/mL) groups. Furthermore, BAEC (Figure 8E) and Her-2 (Figure 8F) demonstrated the cell the migration ability influenced by each treatment. In Figure 8G, the results indicated that each treatment did not inhibit BAEC migration, but promoted the migration distance at 24 h and 48 h. On the contrary, the migration distance of Her-2 cells was significantly decreased in each treatment. However, after treated with Au-Col-BB (10 μg/mL) for 48 h, the migration distance was 0 μm. The above evidence demonstrated after treating with Au-Col-BB (10 μg/mL), the MMP-9 activity in Her-2 cell line was significantly suppressed, leading to the lowest migration distance (Figure 8H). However, no influence in non-transformed BAEC cell, indicating the specific effect of Au-Col-BB.

### 3.7. Assessment of Apoptotic Related Proteins Expression

Western blotting assay was applied to investigate the expression of apoptotic related proteins in BAEC and Her-2 cell lines after treated with different materials for 24 h. BAEC did not express more apoptotic cascade proteins, such as Bax and p21 after culturing with Au-Col-BB (0.5, 1, 5, 10 μg/mL), but was found to significantly expressed after treated with BB (1 μg/mL), which were increased to ~4.5, and ~7.8 fold, respectively (Figure 9A,B).

In line with the quantification results of Her-2 cells, Bax (~6.0 fold) and p21 (~5.1 fold) were observed to be expressed in Au-Col-BB (0.5 μg/mL) group. In addition, the anti-apoptotic protein Bcl-2 and apoptotic resistance protein Cyclin D1, were observed to be suppressed in Au-Col-BB (0.5 μg/mL) group, which were decreased to ~0.2 fold, and ~0.32 fold, respectively (Figure 9C,D). A similar situation could be observed in BB alone (1) group (Bax (~3.8 fold), p21 (~2.9 fold), Bcl-2 (~0.34 fold), Cyclin D1 (~0.35 fold)). The above evidence indicated Au-Col-BB could induce Her-2 cell apoptosis through inducing the expression of Bax and p21, and also decreasing the expression of anti-apoptotic proteins, Bcl-2 and Cyclin D1. Supporting that Au-Col-BB nanocarrier has the capacity of anti-cancer.

### 3.8. Investigation of Tumor Suppression Effects

To validate the anti-tumor capacities of various drugs, different materials were given to the Her-2 cells xenografted mice after tumors reached the appropriate size. After 30 days of treatment, the mice were all sacrificed, and their tumor tissues were also collected for further investigations. The tumor volume was inhibited by Au-Col-BB groups in a dose dependent manner (Figure 10A). During the study period, the body weight of mice in all groups did not change significantly (Figure 10B). The image was shown in Figure 10C, and the weights of tumors based on various groups indicated that Au-Col-BB could suppress the growth of tumors significantly more than Au-Col and control group (Figure 10D). Further, the survival median of the experimental mice of treated with various drugs demonstrated a significantly longer survival treated with Au-Col-BB 20 mg compared to the group treated with other agents (*p* < 0.01) (Figure 10E). Compared with the group of BB only (20 mg), Au-Col--BB 20 mg obviously has better tumor suppressing ability (*p* < 0.001) (Figure 10D), and prolongs the survival time of mice more effectively (*p* < 0.001) (Figure 10E).

## 4. Discussion

Breast cancer is the uncontrolled proliferation of breast cells that leads to malignancy, and commonly occurred among females. Previous studies indicated that the mortality rate of breast cancer can be decreased by early detection, such as mammography, breast self-examination, and clinical breast examination [8]. However, another previous research confirmed that there were several barriers to breast cancer screening, such as poor interaction with doctors or the pain and discomfort caused by the screening procedure [37]. Indeed, surgery and radiation therapies combined with chemotherapy were proved to decrease the risk of recurrent breast cancer. However, the local treatments combined with adjuvant treatments would cause side effects such as myelosuppression, ovarian failure, cardiac toxicity or second cancer [4]. Therefore, creating a novel nanocarrier to delivery anticancer drug with low side effects and high efficiency is a potential method in targeted cancer therapy. Nanoparticles (NPs) such as gold (Au) is a promising candidate due to physicochemical properties including biocompatibility, lower cell toxicity and easily to fabrication with biomolecules. The prior work demonstrates that Au conjugated with biopolymer, such as collagen, polyurethane, and fibronectin had excellent biocompatibility [32,38,39]. Another research also indicated that a polycationic natural polymer, chitosan, combined with Au possessed cytotoxic effect to induce cancer cell death and inhibit clonogenic potential in HeLa and MCF-7 breast cancer cells via ROS generation [40].

Berberine (BB), the natural botanical alkaloid compound, was verified to have multiple therapeutic effects in suppressing tumor growth, reversing drug resistance, and reducing tumor resurgence via regulating various signal pathways [41,42]. In addition, BB also suppressed cell growth through regulating cell cycle, cell autophagy, and enhancing apoptosis [43]. In the current research, a novel collagen gold nanoparticulate nanocarrier conjugated with alkaloid berberine (Au-Col-BB) was successfully prepared and further investigated for biological effects through in vitro and in vivo assay. The MTT assay demonstrated that Au-Col-BB had significantly cytotoxic effect on Her-2 breast cell line compared to the non-transformed BAEC group, especially in the concentration of 10 μg/mL (Figure 3A,B). Literatures indicated that BB could regulate cell cycle and suppress cell growth in various types of cancer diseases. In the A549 lung cancer cell line, BB inhibited the expression of Cyclin D1 leading to G1 phase cycle arrest [44]. Moreover, in HBT-94 chondrosarcoma cells, BB upregulated p53 and p21 expression through activating PI3K/Akt and p38 signaling pathways, leading to G2/M phase arrest [45]. Further, BB could arrest MDA-MB-231 breast cancer cells in S phase to enhance the chemotherapy sensitivity [46]. According to the current evidence (Figure 3D), Her-2 cancer cells were remarkably arrested in Sub-G1 phase in the Au-Col-BB (10 μg/mL) group. An early indicator of apoptosis is the loss of the phospholipid membrane asymmetry of the cell, guiding the exposure of phosphatidylserine on the outer surface of the plasma membrane, and can be investigated through Annexin-V staining [47]. In Figure 4C,D, the amount of Annexin-V positive Her-2 cells was significantly greater in Au-Col-BB (10 μg/mL) group, indicating the early stage of cell apoptosis. Furthermore, berberine was reported to facilitate apoptosis through activating caspases. Literature elucidated that in HL-60 leukemia cells, BB promoted cell apoptosis by increasing caspase-8/9 expression and caspase-3 activation, then leading to Bcl-2 inhibition [48]. In addition, mitochondria were verified to be critical in regulating apoptosis. Previous research indicated the activation of caspases stimulated cell apoptosis through increasing the permeability of mitochondria membrane [49]. BB also increased the phosphorylation of p53 to enhance the apoptotic protein Bax entry into mitochondria [50]. In line with the previous research, our results demonstrated that after treating Her-2 breast cancer cells with Au-Col-BB, the apoptotic cascade proteins Bax and p21 were significantly induced to be overexpressed, particularly in the Au-Col-BB (0.5 μg/mL group. BB was also reported to suppress migration by targeting ephrin-B2 then decreased MMP-2/9 protein expression in breast cancer ZR-75-30 cells [51]. BB inhibited MMP-2/9 expression through downregulation of TGF-β1, suppressing proliferation and movement of triple negative breast cancer cells (TNBC) [52]. Results in current research demonstrated that the expression of MMP-9 in Her-2 cell line was remarkably decreased in the Au-Col-BB (10 μg/mL) group, leading to the least distance of cell moving distance at 48 h (Figure 8). Based on the above evidences, the novel Au-Col-BB nanocarrier exhibited the strong ability to suppress cell proliferation via regulating cell cycle progression, inducing apoptosis through various mechanisms, and affecting the expression of MMPs to inhibit cell invasion and metastasis.

Endocytosis is the major route for cell entry of nanoparticles associated with the size, surface charge, and surface properties [53]. Based on the excellent cellular uptake ability, Au-Col-BB was evaluated for the ability to mediate the cell uptake mechanisms in the non-transformed BAEC and Her-2 breast cancer cell lines. Several pharmaceutic inhibitors were used to determine which endocytic mechanisms involved in the internalization of Au-Col-BB nanoparticles, Cytochalasin D (Cyto-D), Chlorpromazine (CPZ), Bafilomycin (Baf), and 2-Mercaptoethanol (β-ME). Cyto-D is a cell-permeable and potent inhibitor for actin polymerization and have been proved to disturb micropinocytosis [54], CPZ inhibits clathrin-mediated endocytosis through anchoring clathrin and adaptor protein 2 (AP2) complex to endosomes and preventing the assembly of coated pits at the inner surface of plasma membrane [55], Baf was reported to suppress vacuolar ATPase inhibitor that was associated with interfering cell autophagy [56], and β-ME inhibited phagocytosis. Our results indicated that internalization of Au-Col-BB was significantly suppressed by CPZ and Baf, suggesting that clathrin-mediated endocytosis and cell autophagy are the main pathways for Au-Col-BB entry into cells (Figure 7). The results supported that cell autophagy was the main internalization mechanism for Au-Col-BB nanocarrier by lysosome (Figure 6D–F). In addition, the size of Au-Col-BB, 227 nm, may enhance permeability and retension effects to the cells. Previous studies proved that the nanoparticles with a size range of 10 to 1000 nm could enter into cell vesicles. For detail, the large particles in 200 to 500 nm may be internalized via micropinocytosis, those around 10 to 300 nm and 60 to 80 nm would undergo clathrin-mediated endocytosis and caveolae-mediated endocytosis, while particles size lower than 100 nm processed clathrin-independent pathway [57,58].

Various NPs were explored and applied to target tumor cells precisely for avoiding any risk to harm normal cells or organs. NPs-mediated targeted drug delivery systems are suggested to be a novel technique to treat carcinoma [59]. One literature indicated that berberine combined with AgNPs (Ag-BB) enhanced the development of oxidative stress in Ehrlich solid carcinoma (ESC) tissue by increasing nitric oxide (NO), antioxidant proteins (glutathione), and lipid peroxidation (LPO). Ag-BB was also found to stimulate apoptotic cascade expression in tumor cells through downregulating of anti-apoptotic protein (Bcl-2) expression and upregulating the mRNA expression of pro-apoptotic proteins (Bax and caspase-3) [60]. Above research provided a novel approach for future investigations in the AgNPs delivery system. In a recent study, berberine was encapsulated in bovine serum albumin (BSA) nanoparticles (BB-BSA NPs) as a novel nanocarrier for breast cancer treatment. BB-BSA NPs demonstrated the superior anticancer activity compared to BB alone against MDA-MB-231 breast cancer cells. Further, assessments of cell apoptosis and uptake ability also verified BB-BSA NPs to be more cytotoxic and higher intracellular uptake towards breast cancer cells. However, the stability results suggested that BB-BSA NPs were considerably stable in pH 7.4 aqueous solution, but under acidic condition (pH 5) would degraded [61]. Lysosomes are membrane-bound organelles which are major in cellular processes, such as endocytosis, phagocytosis, and autophagy to maintain cellular homeostasis through creating an acidic environment (pH 4.5–5.0) and recruiting hydrolytic enzymes that degrade engulfed biomolecules [62]. Based on our evidence from Lysotracker assay, Au-Col-BB nanocarrier would not degrade under acidic condition and was stable, suggesting Au-Col-BB to be a potential nanodrug for targeting Her-2 breast carcinoma.

This work explored the novel collagen gold nanoparticles combined with berberine (Au-Col-BB) as a potential cancer drug, by evaluation of physicochemical properties, cell toxicity, cell physiology, uptake mechanisms, and in vivo xenograft mice model. Au-Col-BB were selectively toxic towards Her-2 breast cancer cells but not non-transformed BAEC. Au-Col-BB induced cancer cell death more efficiently especially in 10 μg/mL compared to pure berberine. Further, Au-Col-BB significantly inhibited cell proliferation, and induced apoptotic cascade protein (Bax and p21) overexpression to promote cell apoptosis. The cellular uptake assessments indicated that Au-Col-BB nanocarriers were therapeutically more effective and improve the anticancer capacity of pure berberine through target site delivery. The evidences from in vivo xenograft mice strongly suggested that Au-Col-BB nano drugs remarkably decreased the tumor weight and volume of the experimental animals, and did not exhibit any side effects such as body weight loss. Therefore, Au-Col combined with the natural berberine may be become a novel candidate to improve the quality and efficiency for targeting breast cancer treatments.

## 5. Conclusions

A novel derived gold nanoparticle-collagen nanocarrier incorporated with alkaloid berberine (Au-Col-BB) was prepared and investigated the physicochemical properties and biological effects on BAEC and Her-2 breast cancer cell lines. The average diameter of Au-Col-BB was approximately 227 nm. Au-Col-BB demonstrated remarkable cytotoxicity to induce cell apoptosis in Her-2 cell lines. Meanwhile, Au-Col-BB exhibited nontoxicity and superior biocompatibility to non-transformed BAEC. Moreover, the nanocarrier Au-Col-BB had better cellular uptake ability than Au-Col in both Her-2 and BAEC cells. The analysis of uptake mechanisms revealed that cell autophagy and clathrin-mediated endocytosis were the major mechanisms for Au-Col-BB internalization in both cell types. Furthermore, the MMP activity and cell migration assay demonstrated that Au-Col-BB (10 μg/mL) significantly inhibited cell movement in Her-2 cell line. Moreover, the Western blot analysis indicated that Au-Col-BB induced apoptotic related proteins, such as Bax and p21, to be expressed. Particularly, the mice xenograft model demonstrated that Au-Col-BB nanocarrier effectively reduced the tumor weight and increased the animal survival period. In conclusion, this study validates the antitumor effect and the endocytic mechanisms of Au-Col-BB nanocarrier in Her-2 breast carcinoma cell lines and non-transformed BAEC. These promising findings offer an innovative approach in enhancing the cellular uptake of nutraceuticals for cancer treatment and nutrition therapy.

## Figures and Tables

**Figure 1 cancers-13-05317-f001:**
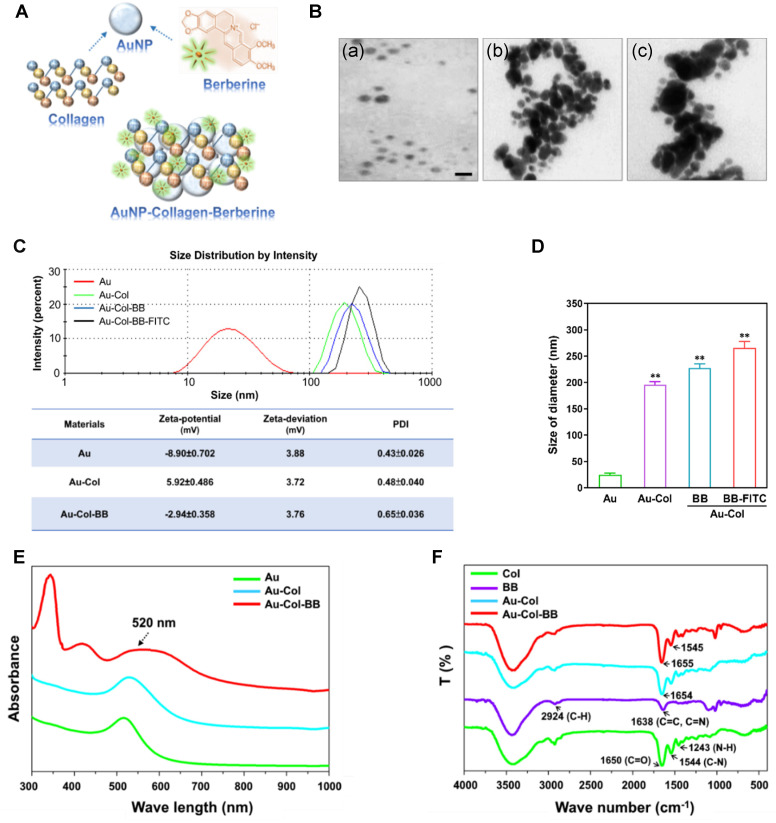
Material characterization of Au-Col derived nanocomposites conjugated with berberine. (**A**) Schematic illustra-tion for preparing gold nanoparticle-collagen-berberine (Au-Col-BB) composites. Collagen was conjugated with nano gold (Au) by sonication, then Au-Col was further cross-linked with berberine at 3:2 volume ratio to obtain Au-Col-BB. (**B**) TEM image of (**a**) Au, (**b**) Au-Col, and (**c**) Au-Col-BB. Scale bar = 20 nm. (**C**) The size distribution intensity of AuNP-derived nanocarrier were determined by using DLS. Surface charge of Au, Au-Col, and Au-Col-BB was further investigated with Zeta-potential, Zeta-deviation, as well as polydispersity index (PDI) value. The Zeta-potential of each material was −9.7, 6.88, and −2.94 mV; the value of Zeta-deviation was 3.92, 3.87, and 3.76 mV: the PDI value was 0.41, 0.49, and 0.58, respect-tively. (**D**) The diameter measured by DLS assay was 24.5, 196, 227, 265 nm for Au, Au-Col, Au-Col-BB, and Au-Col-BB-FITC, respectively. The results are representative of one of three independent experiments. ** *p* < 0.01: greater than the control. (**E**) UV-Visible spectra confirmed each nanomaterial containing AuNPs with the typical absorption peak at 520 nm. (**F**) The FTIR spectra of various materials in total wave number ranges from 400 cm^−1^ to 4000 cm^−1^.

**Figure 2 cancers-13-05317-f002:**
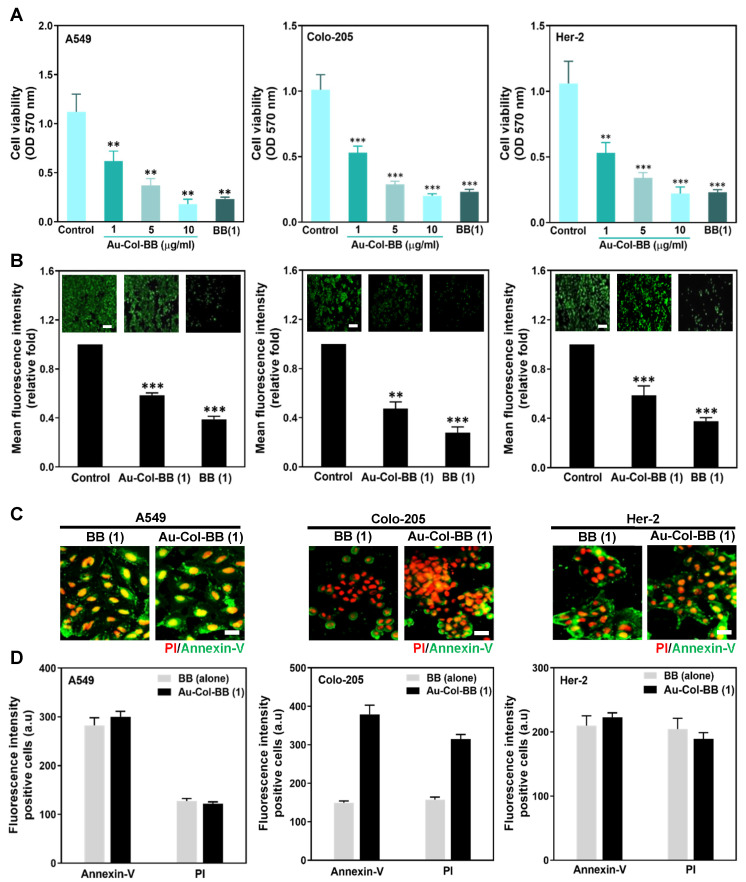
Biocompatibility assay for Au-Col-BB between A549, Colo-205, and Her-2 cell lines. (**A**) The cytotoxicity between A549, Colo-205, and Her-2 cell lines seeded on various materials was investigated by MTT assay. The cell growth was observed with significantly inhibition of each cell line especially for Au-Col-BB (10 μg/mL) and BB (1 μg/mL) at 48 h. (**B**) Calcein-AM staining (green color fluorescence) was also applied to investigate cell viability in Au-Col-BB (1 μg/mL) and BB (1 μg/mL) at 48 h. Semi-quantitative data based on fluorescence intensity shared similar results with MTT assay. Scale bar = 50 μm. (**C**,**D**) Each cell line was subjected to BB (1 μg/mL) and Au-Col-BB (1 μg/mL) for 48 h, then double stained by Annexin V (green) -PI (red) for fluorescence microscopy and flow cytometry investigation. In Her-2 cell line, both BB (1 μg/mL) and Au-Col-BB (1 μg/mL) could induce more apoptotic (Annexin V^+^) and dead (PI^+^) cells. Scale bars = 50 μm. ** *p* < 0.01, *****
*p* < 0.001: compared to the control group.

**Figure 3 cancers-13-05317-f003:**
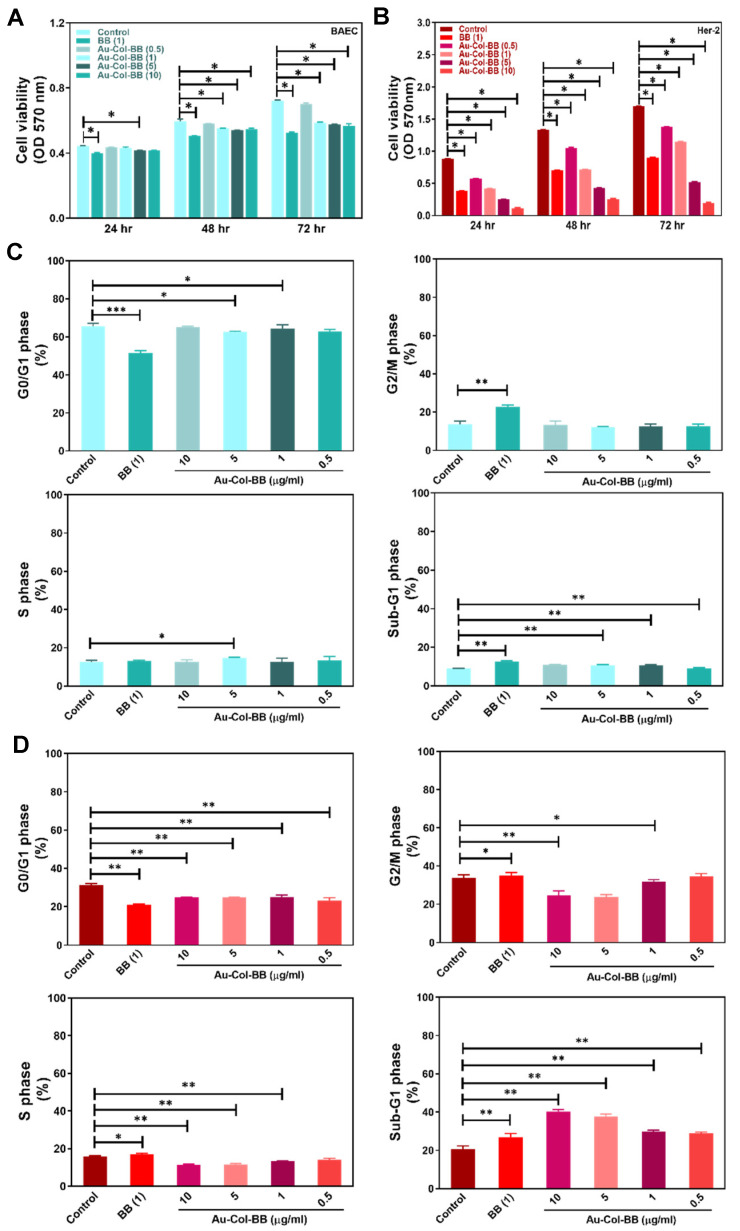
Biocompatibility assay for Au-Col-BB between BAEC and Her-2 cell line. BAEC and Her-2 cell line was exposed to BB (1 μg/mL) and Au-Col-BB containing various concentration of BB (0.5, 1, 5, 10 μg/mL). (**A**,**B**) The cell viability was measured at 24, 48, and 72 h. Cell growth of BAEC was increased at 48 and 72 h seeded on different materials. In contrast, the growth of Her-2 cell line was significantly inhibited, particularly in Au-Col-BB (10 μg/mL) group at 24, 48, and 72 h. Additionally, cell cycle influenced by various materials was also analyzed. (**C**) G0/G1 population of BAEC was significantly reduced in BB group (1 μg/mL), but slightly decreased in Au-Col-BB groups. Additionally, the cell population of G0/G1 phase was more than G2/M, S and Sub-G1 phase. (**D**) Cell populations of Her-2 cell line at sub-G1 phase was obviously increased especially in Au-Col-BB (10 μg/mL) group. Additionally, the population of G0/G1 phased was significantly decreased after various treatments. * *p* < 0.05, ** *p* < 0.01, *** *p* < 0.001: compared to the control group.

**Figure 4 cancers-13-05317-f004:**
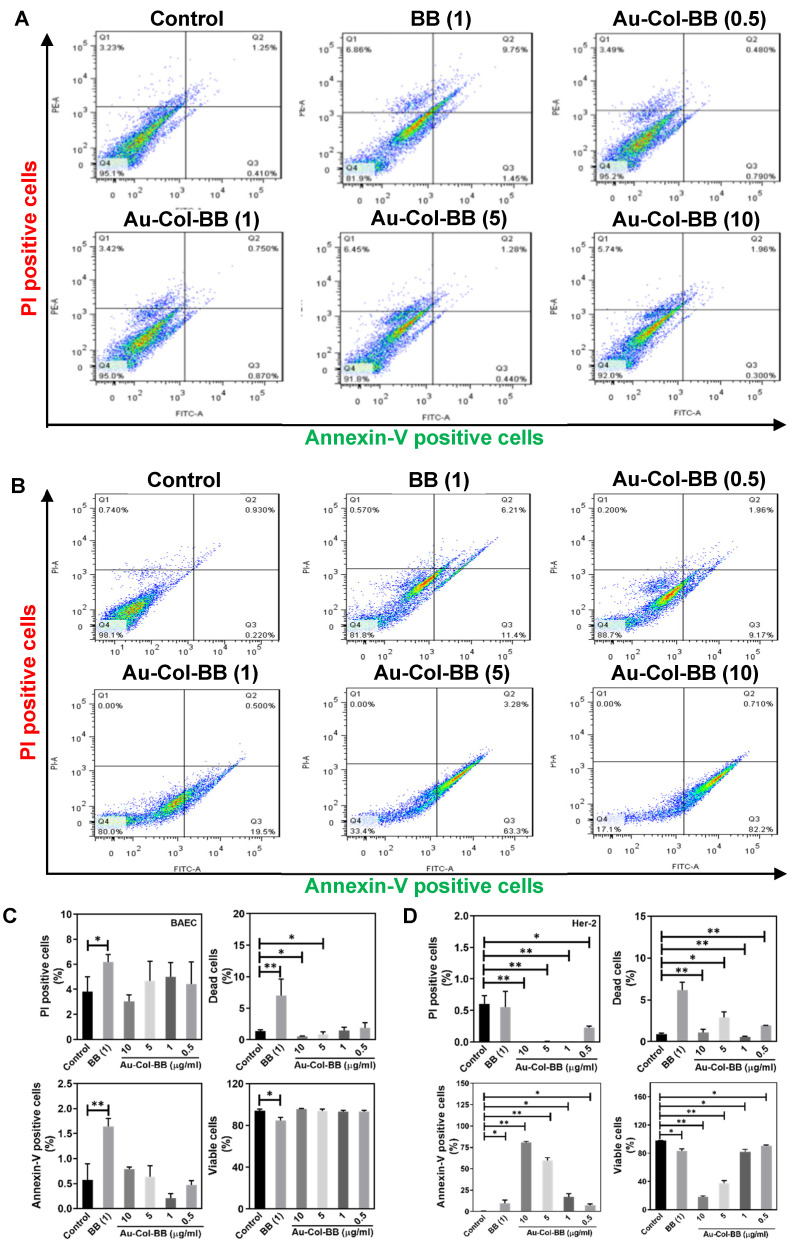
Annexin V-PI double staining to detect apoptotic BAEC and Her-2 cells by using flowcytometry at 48 h. The cell population of (**A**) BAEC and (**B**) Her-2 cells was demonstrated. (**C**) BAEC appeared to be not injured by Au-Col-BB in terms of apoptotic and dead cell amount. However, while incubating with BB (1 μg/mL), the amount of Annexin-V positive cell and PI positive cell was 1.64% and 6.19%, and the dead cell amount was figured out for 6.99%. The quantification of viable BAEC cells also showed high survival ratio in each Au-Col-BB group, but slightly increased in BB (1 μg/mL). (**D**) The quantitative results evaluated that the amount of apoptotic Her-2 cells (Annexin-V positive) was remarkably increased, especially in Au-Col-BB (10 μg/mL), which was figured out for 80.6%. The amounts of dead cells induced by BB (1 μg/mL) was 6.15% that significantly more than control and other groups. The quantification of viable Her-2 cells also showed lowest survival ratio was 18.2% in Au-Col-BB (10 μg/mL) group. * *p* < 0.05, ** *p* < 0.01: compared to the control group.

**Figure 5 cancers-13-05317-f005:**
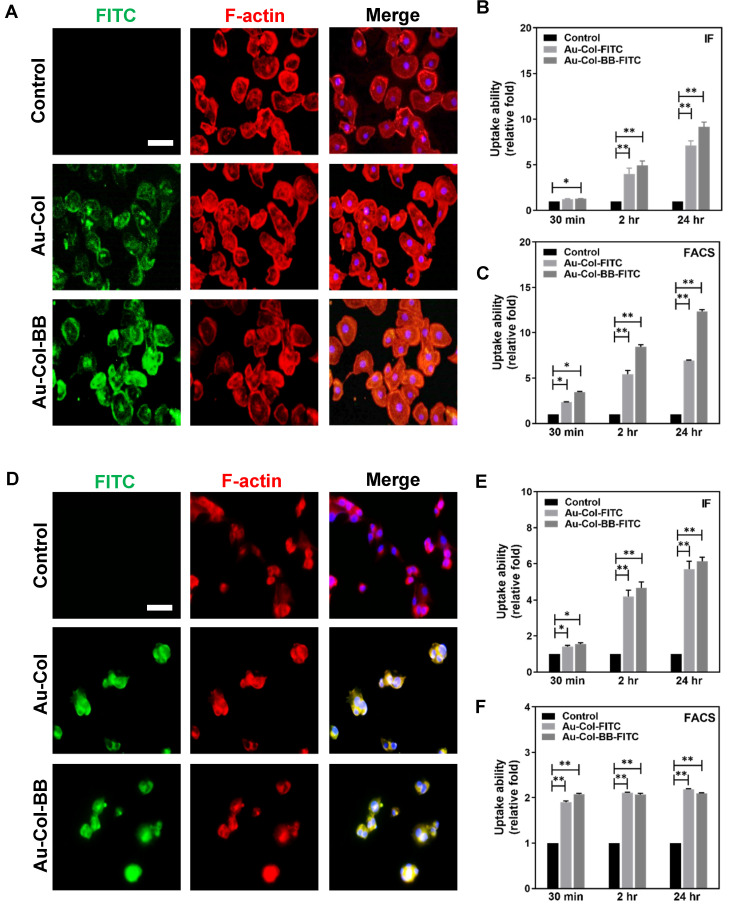
Assessment of cell uptake ability in BAEC and Her-2 cell line. Au-Col and Au-Col-BB were firstly conjugated with fluorescent dye (FITC) to investigate inside cell transportation, then observed by using fluorescent microscopy. The immunofluorescence (IF) images at 2 h were displayed as (**A**) BAEC and (**D**) Her-2 cells. The semi-quantified uptake ability of BAEC was demonstrated through (**B**) IF and (**C**) fluorescence activated cell sorter (FACS) method. The results from IF and FACS both indicated the uptake amount of Au-Col-BB was significantly increased at 2 and 24 h. In addition, in Her-2 breast cancer cell line, the uptake efficiency was also quantified based on (**E**) IF and (**F**) FACS method. The IF results also indicated the uptake amount of Au-Col-BB was the greatest in Her-2 cells at 2 and 24 h. The quantified data from FACS showed the uptake amount of Au-Col-BB and Au-Col was higher than the control at each time point. The concentration of Au-Col-BB was 1 ug/mL for the experiments. Scale bars = 20 μm. * *p* < 0.05, ** *p* < 0.01: compared to the control group.

**Figure 6 cancers-13-05317-f006:**
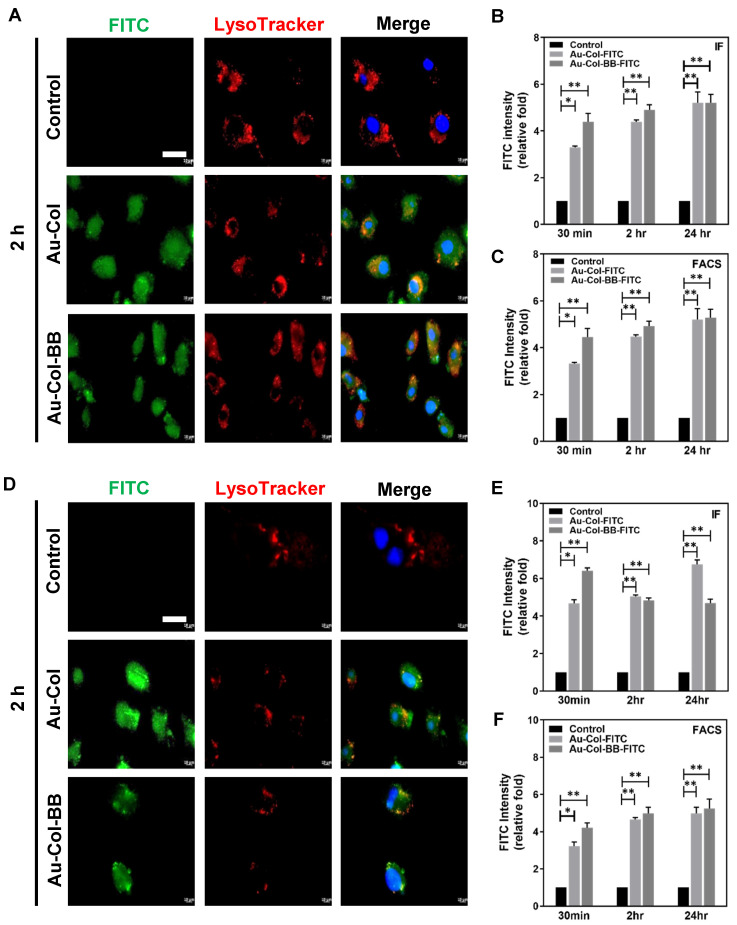
Evaluation of cell uptake ability in BAEC and Her-2 cells line by using Lysotracker. A red fluorescent probe, Lysotracker was applied to observe lysosomes so as to verify potential transportation ability, and the lysosomes were observed within cytoplasm, particularly perinuclear site. (**A**) Non-transformed BAEC cell line was also treated with Au-Col and Au-Col-BB flagged with FITC for 2 h, both materials (green) were co-localized with lysosomes (red). (**B**) The uptake efficiency quantified by fluorescent intensity of Au-Col group at 30 min was ~3.3 fold, then increased to ~4.4 and ~5.2 fold at 2 h and 24 h. The Au-Col-BB group also shared similar trend. Both were indicated for an increased trend. (**C**) The uptake amount of BAEC was measured by FACS method. The uptake amount for Au-Col-BB increased to ~4.92 and ~5.28 fold at 2 h and 24 h compared to 30 min (~4.46 fold). The similar trend was also obderved in Au-Col group (30 min: ~3.32 fold, 2 h: ~4.48 fold, 24 h: ~5.2 fold). (**D**) The images of Her-2 cells showed after treating with Au-Col-FITC and Au-Col-BB-FITC for 2 h, both materials (green) were colocalized with lysosomes (red), (**E**) The quantification results of IF method indicated that the uptake ability of Au-Col-BB at 30 min was ~6.41 fold, then decreased to ~4.84 and ~4.69 fold at 2 and 24 h. In contrast, Au-Col group showed increased at 2 h and 24 h, which was figured out as ~5.08 and ~6.76 fold. (**F**) To justify potential observation bias from immunofluorescence, FACS method was also applied. The uptake amount in Au-Col-BB group increased to ~4.98 and ~5.25 fold at 2 h and 24 h, compared to 30 min (~4.22 fold). Likewise, the similar condition was observed for Au-Col group (30 min: ~3.22 fold, 2 h: ~4.65 fold, 24 h: ~4.98 fold). The concentration of Au-Col-BB was 1 ug/mL for the experiments. Scale bars = 10 μm. * *p* < 0.05, ** *p* < 0.01: compared to the control group.

**Figure 7 cancers-13-05317-f007:**
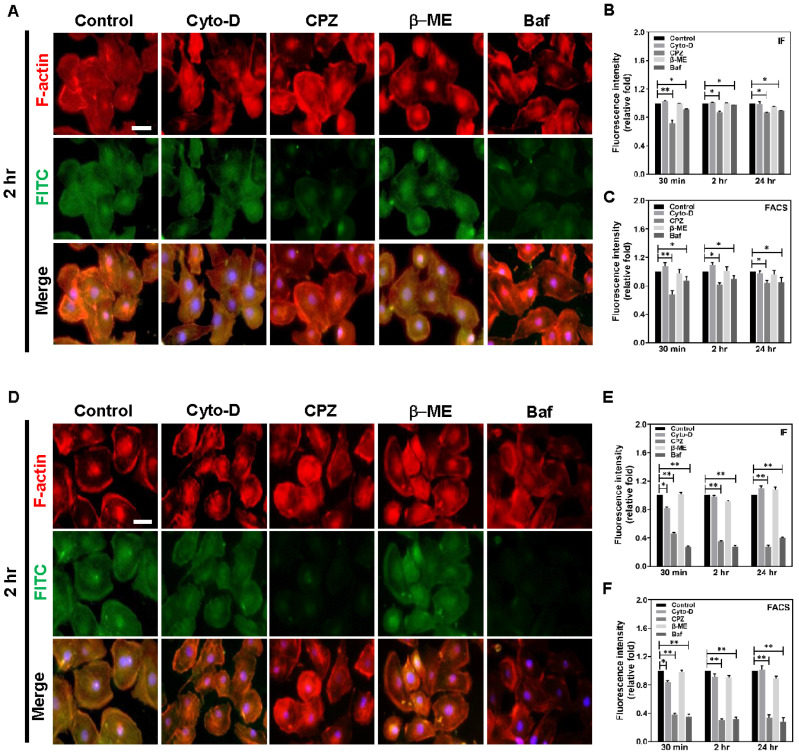
Assessment of endocytotic route for BAEC and Her-2 cell lines were also treated with 4 kinds of endocytosis inhibitors. There are several different types of endocytotic pathways: caveolae, macropinocytosis, receptor-mediated en-docytosis, and phagocytosis. 4 different endocytotic inhibitors, Cytochalasin D (Cyto-D), Chlorpromazine (CPZ), 2-Mercaptoethanol (β-ME), and Bafilomycin (Baf) were incubated with Au-Col-FITC-BB (10 μg/mL) for various times (30 min, 2 h, and 24 h). (**A**) The fluorescence images of BAEC at 2 h. (**B**) The quantitative results of fluorescence intensity also indicated the colocalization of Au-Col-BB with F-actin was remarkably decreased after treating with CPZ and Baf at 30 min, 2 h and 24 h. (**C**) The fluorescein positive cells were quantified by FACS method, and also demonstrated the decreased intensity influenced by CPZ and Baf at each time point. (**D**) The colocalization of Au-Col-BB in Her-2 cell line was signif-icantly decreased after treating CPZ and Baf inhibitors at 2 h. Fluorescent intensity of Au-Col-BB measured by (**E**) IF method indicated that the fluorescence density of Au-Col-BB was obviously decreased in CPZ and Baf at each time point. (**F**) FACS method was also applied to avoid selection bias from IF method, and the quantification results were similar to IF methods. These findings indicate diverse routes present for endocytosis mediated Au-Col-BB entrance. The concentra-tion of Au-Col-BB was 1 ug/mL for the experiments. Data were presented as the mean ± SD (*n* = 3). The scale bar equals to 20 μm.* *p* < 0.05, ** *p* < 0.01: compared to the control group.

**Figure 8 cancers-13-05317-f008:**
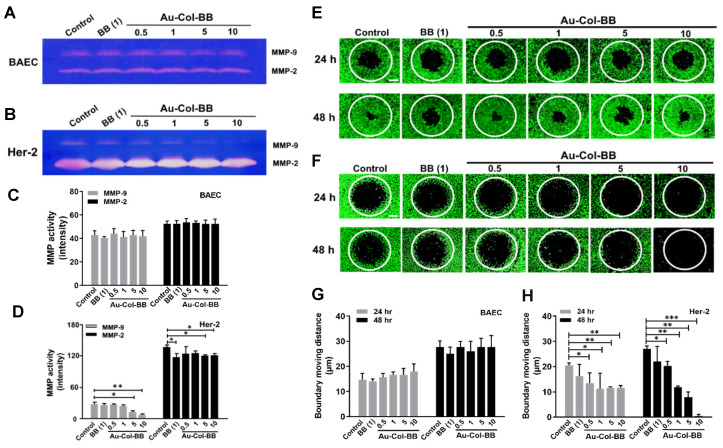
The MMP enzymatic activities in non-transformed BAEC and Her-2 cancer cell culturing with different materials at 48 h. The representative zymogram of (**A**) BAEC and (**B**) Her-2 for MMP-2 and MMP-9 at 48 h is shown to explore the potential mechanism involved with cell migration. (**C**) The expression of MMP-2 and 9 in BAEC did not show any dispar-ity in various materials. (**D**) In Her-2 cell line, matrix metalloproteinases MMP-9 were significantly reduced after treated with Au-Col-BB, especially at the concentration of 10 μg/mL. Meanwhile, the quantification of MMP-2 was slightly de-creased in each group compared to control. * *p* < 0.05, ** *p* < 0.01. Observation of migration ability in BAEC and Her-2 cell lines after Au-Col-BB treatment. The migration cells were observed by using microscopy. (**E**,**G**) By contrast, the migration ability of non-transformed BAEC was increased after treating with different materials at 48 h. (**F**,**H**) The fluorescence im-ages indicated that the migration ability of Her-2 cancer cells was evidently decreased. Likewise, the quantification results measured by fluorescence intensity figured out the migration distance of Her-2 cell line were significantly decreased, especially treated with Au-Col-BB (10 μg/mL) at 48 h. Data are presented as mean ± SD (*n* = 3). Scale bars = 50 μm. * *p* < 0.05, ** *p* < 0.01, *** *p* < 0.001: compared to the control group.

**Figure 9 cancers-13-05317-f009:**
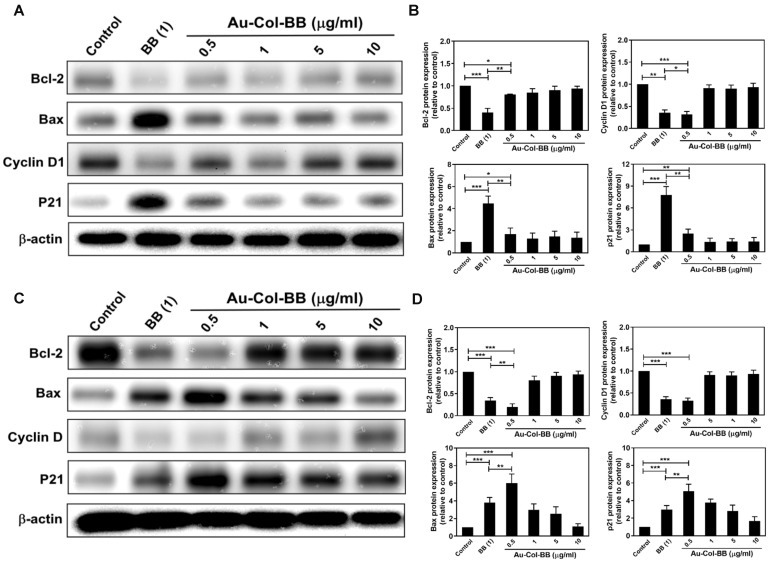
Assessment of apoptosis related cascade proteins in non-transformed BAEC and Her-2 cell line after treated with various materials for 24 h. (**A**) The representative western blot images of BAEC were demonstrated. (**B**) The quantification results indicated that the anti-apoptotic protein Bcl-2 were significantly inhibited especially in BB (1 μg/mL) group (~0.4 fold), but slightly decreased in Au-Col-BB (0.5 μg/mL) group (~0.8 fold). In addition, the expression level of apoptotic resistance protein Cyclin D1 was decreased in both BB (1 μg/mL) and Au-Col-BB (0.5 μg/mL) group with the value of ~0.35 and ~0.32 fold. Otherwise, the pro-apoptotic protein Bax and anti-proliferation protein p21, were also demonstrated remarkably increased especially in BB (1 μg/mL) group, and the expression level was ~4.5 and ~7.8 fold. In Au-Col-BB (0.5 μg/mL) group evaluated less expression of Bax and p21 (~1.7 and ~2.5 fold), respectively. (**C**) The representative western blot images of Her-2 cells were shown. (**D**) Based on the quantification results measured by image J, the anti-apoptotic protein Bcl-2 and apoptotic resistance protein Cyclin D1, were significantly inhibited especially in Au-Col-BB (0.5 μg/mL) and BB (1 μg/mL) group. Where the expression of Bcl-2 and Cyclin D1 in Au-Col-BB group was decreased to ~0.2 and ~0.32 fold, and culturing with BB (1 μg/mL~0.34 and ~0.35 fold. Otherwise, the proapoptotic protein Bax and anti-proli-feration protein p21, were also determined and showed significantly increased in Au-Col-BB (0.5 μg/mL) group, the ex-pression level was observed to be ~6.0 and ~5.1 fold, respectively. * *p* < 0.05, ** *p* < 0.01, *** *p* < 0.001.

**Figure 10 cancers-13-05317-f010:**
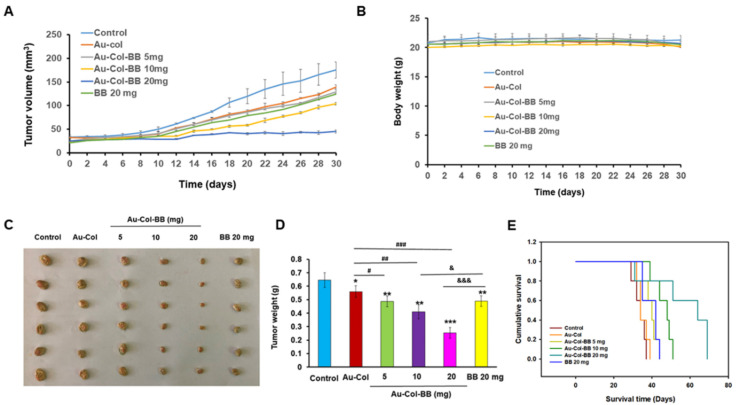
Assessment of Au-Col-BB treatment effects in mice model. Control Au-Col-BB demonstrated an excellent anti-tumor performance in Her-2 tumor xenografts. After injection of Her-2 cells, tumor-forming nude mice were randomly assigned (*n* = 6 in each group) to receive different materials. The control group was the mice treated with PBS. (**A**) Tumor volume of the experimental mice receiving various treatments is shown. Tumors were remarkably smaller in mice after treating with Au-Col-BB 20 mg compared to the mice treated with control (PBS), Au-Col, and BB only 20 mg. (**B**) Body weight curve of the experimental mice treating with various drugs is shown. The body weight was not remarkably changed in mice treated with different materials. (**C**) The images of tumors taken from the mice sacrificed at the end of the study. (**D**) Tumor weight of the experimental mice treating with various drugs is shown. Tumors were remarkably lighter in mice treated with the Au-Col-BB 20 mg than in mice treated control (PBS), Au-Col, and BB only 20 mg (* *p* < 0.05, ** *p* < 0.01, *** *p* < 0.001). (^#^
*p* < 0.05, ^##^
*p* < 0.01, ^###^
*p* < 0.001: compared to Au-Col) (^&^
*p* < 0.05, ^&&&^
*p* < 0.001: compared to BB 20 mg). (**E**) The survival median of various treatment mice groups (*n* = 5) was evaluated up to day 70. The median survival of mice was significantly longer injected with Au-Col-BB 20 mg evaluated by Kaplan–Meier statistics and log-rank test.

## Data Availability

Data are contained within the article.

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
