# Peer review of "Delivery Capacity and Anticancer Ability of the Berberine-Loaded Gold Nanoparticles to Promote the Apoptosis Effect in Breast Cancer"

_cancers, 2021, doi:10.3390/cancers13215317_

Round 1
Reviewer 1 Report
It is an interesting study to investigate anti-cancer activity of berberine-loaded gold nanoparticles in breast cancer. Authors also try to figure out possible mechanisms.
The suggestions/comments are given below.
- Authors need to check all the data carefully. There are a lot of mislabeling in figures and figure legends. For example, Figure 8H, unit of y-axis is not correct; the legend’s description of Figure 9A&C is wrong; Figure 7B shows fluorescence images, but not quantitative results; Authors do not label (D) in figure 6.
- Authors need to confirm the quantitative results are consistent with the image data. For example, Figure 8 E&G and Figure 8 F&H, especially in Figure 8 F&H, readers can’t find migration activity is inhibited by different does of Au-Col-BB. Meanwhile, authors should explain the cell density is not consistent in figure 8F, especially at the concentration of 0.5 μg/ml Au-Col-BB.
- Authors need to check the Statistics analysis is correct or not. For example, Figure 8H, it may me confuse why the treatment of 0.5 μg/ml Au-Col-BB shows higher statistically significance than the treatment of 10 μg/ml Au-Col-BB, when compared to control group. A similar situation in Figure 9C.
- The description in section 3.5. and legend of Figure 6 is not correct. Authors described in Line 530, (E) The quantification results of IF method indicated that the uptake ability of Au-Col-BB at 30 min was 6.41 fold, then decreased to 4.84 and 4.69 fold at 2 and 24 hr. In contrast, Au-Col group showed increased at 2 hr and 24 hr, which was figured out as 5.08 and 6.76 fold”, but the description should be indicated (B) in Figure 6.
- Authors do not clearly describe materials and methods in Tumor xenograft mouse model about the survival median of the mice.
Author Response
Reviewer 1:
It is an interesting study to investigate anti-cancer activity of berberine-loaded gold nanoparticles in breast cancer. Authors also try to figure out possible mechanisms.
The suggestions/comments are given below.
- Authors need to check all the data carefully. There are a lot of mislabeling in figures and figure legends. For example, Figure 8H, unit of y-axis is not correct; the legend’s description of Figure 9A&C is wrong; Figure 7B shows fluorescence images, but not quantitative results; Authors do not label (D) in figure 6.
Answer:
Thanks for the suggestion from the reviewer.
(1) We have modified the unit of y-axis in Figure 8H. (Page 19)
(2) We have modified the legend’s description of Figure 9A&C. (Page 20)
(3) We have changed the label (B) to the correct figure in Figure 7. (Page 18)
(4) We have labeled (D) in Figure 6. (Page 16)
- Authors need to confirm the quantitative results are consistent with the image data. For example, Figure 8 E&G and Figure 8 F&H, especially in Figure 8 F&H, readers can’t find migration activity is inhibited by different does of Au-Col-BB. Meanwhile, authors should explain the cell density is not consistent in figure 8F, especially at the concentration of 0.5 μg/ml Au-Col-BB.
Answer:
Thanks for the suggestion from the reviewer.
(1) We have modified the figure legend of Figure 8F to correspond to Figure 8H. (Page 19)
(2) We have corrected the picture and changed the labeling in each group make it to be consistent.
- Authors need to check the Statistics analysis is correct or not. For example, Figure 8H, it may me confuse why the treatment of 0.5 μg/ml Au-Col-BB shows higher statistically significance than the treatment of 10 μg/ml Au-Col-BB, when compared to control group. A similar situation in Figure 9C.
Answer:
Thanks for the suggestion from the Reviewer. We have checked the statistics analysis and modified the data in Figure 8H and Figure 9C. (Page 19 and Page 20)
- The description in section 3.5. and legend of Figure 6 is not correct. Authors described in Line 530, (E) The quantification results of IF method indicated that the uptake ability of Au-Col-BB at 30 min was 6.41 fold, then decreased to 4.84 and 4.69 fold at 2 and 24 hr. In contrast, Au-Col group showed increased at 2 hr and 24 hr, which was figured out as 5.08 and 6.76 fold”, but the description should be indicated (B) in Figure 6.
Answer:
We have carefully checked the quantitative data in Figure 6B, 6C, 6E, 6F and make sure the figures are corresponded to Results description and Figure legend.
- Authors do not clearly describe materials and methods in Tumor xenograft mouse model about the survival median of the mice.
Answer:
Thanks for the suggestion from the Reviewer. I have clearly described materials and methods in Tumor xenograft mouse model including the survival median of the mice (Page 7-8, line 347-363).
Reviewer 2 Report
The authors describe a novel way of delivering BB to as an anti-cancer therapeutic using Au-Col NPs, highlighting that the Au-Col-NPs has higher biocompatibility and more cytotoxic ability, compared to BB alone. The authors also show that Au-Col NPs has cytotoxic effects in vitro and in vivo, ss well as ability to reduce cancer cells migration. While this is an interesting study, some of the results presented do not strongly support the authors’ proposition, such as in Fig. 2A, where BB shows higher cytotoxic effects at a relatively lower concentration (1 ug/mL) compared to up to 10 ug/mL of Au-Col-BB NPs. Hence, this makes one wonder what is the added advantage of using Au-Col-BB NPs as a delivery platform than just using BB alone?
Comments:
1) Fig. 1B and 1C – polydispersity index of the nanoparticles are too high – usually for a monodisperse population, the PDI should be < 0.1. As a result, how does the authors ensure that the effect of Au-Col-BB NPs is homogenous for all the assays?
2) Fig. 2A – Why is the 10 ug/mL Au-Col-BB treated cells having a higher viability compared to 1 ug/mL and 5 ugmL? Shouldn’t there be a dose dependent response? For Fig. 2B, BB only is able to give a greater cytotoxic effect compared to Au-Col-BB NPs – why is using Au-Col-BB NPs more advantageous for further applications?
3) For Fig. 3B, there is a lack of dose dependent response for the Au-Col-BB NPs – even with a concentration increase of 20 times (from 0.5 to 10 ug/mL), the cell viability is about the same. Is there any explanation for this? How does this inform the optimal concentration to use for subsequent assays? In Fig. 3C the labeling of Au-Col-BB is different than the other panels.
4) Based on the in vitro assays, it looks like BB already exerts a strong effect on killing cancer cells, comparable to Au-Col-BB NPs. However, this control group was not included in the mice work. Hence, it makes one wonder if the Au-Col-BB NPs indeed have better effects compared to BB, or is treatment with only BB enough (i.e., adding into drinking water, oral gavage etc)? It will be a stronger argument if there are other data supporting that either Au-Col-BB NPs are able to be more biocompatible (i.e., less inflammation, less toxicity to surrounding tissues than BB), or that Au-Col-BB NPs have a higher localization effect or longer circulation times than BB only.
5) In Fig. 10E, it seems that the survival time for Au-Col-BB 5 mg is even lesser than control group? Does this indicate any signs of toxicity that Au-Col-BB might be causing?
6) It is unclear from the material and methods section, as well as in the manuscript, how are the NPs being administered to the mice? It seems from Fig. 11 that the NPs are being treated via intratumor injection? Please provide clarity on this method in the appropriate sections and descriptions. Assuming it is indeed an intratumor injection, it is a highly localized injection at the tumor site. Hence, the significant effects in tumor volume reduction do not seem too surprising to me. One of the advantages of NPs is to improve blood circulation times and specificity of targeting to tumors due to the EPR effect. Hence, some common route of injections are intravenous and intraperitoneal. Why are these two routes not being used instead of intratumor? Will the Au-Col-BB NPs achieve the same extent of effect if it was done by either of these two routes?
Other minor comments:
- There are a lot of language errors including incoherent sentence structures, grammar mistakes and typos. For instance, a lot of sentences start with “Then the…” instead of using conjunctions such as “Next, we did this and that”; or “Additionally, we saw this and that”. The authors should improve on the language and proof-reading.
- For the results section, there was very few data interpretation. The results are just described literally as how they are presented in the figures, without explaining clearly the rationale of doing the assays, and why the results make sense or are significant in proving a certain point/hypothesis.
- 6 should be LysoTracker dye, not Lysotracer dye. Fig. 6D labeling is missing.
- 11 resolution is fuzzy.
- The statistics labeling for the figures are confusing. For example, in Fig. 3A and 3B, it is not clear what are the two treatment groups that are being compared and which * corresponds to which comparison. Please indicate the statistics clearly and consistently for each figure.
Author Response
Reviewer 2:
The authors describe a novel way of delivering BB to as an anti-cancer therapeutic using Au-Col NPs, highlighting that the Au-Col-NPs has higher biocompatibility and more cytotoxic ability, compared to BB alone. The authors also show that Au-Col NPs has cytotoxic effects in vitro and in vivo, ss well as ability to reduce cancer cells migration. While this is an interesting study, some of the results presented do not strongly support the authors’ proposition, such as in Fig. 2A, where BB shows higher cytotoxic effects at a relatively lower concentration (1 ug/mL) compared to up to 10 ug/mL of Au-Col-BB NPs. Hence, this makes one wonder what is the added advantage of using Au-Col-BB NPs as a delivery platform than just using BB alone?
Comments:
- Fig. 1B and 1C – polydispersity index of the nanoparticles are too high – usually for a monodisperse population, the PDI should be < 0.1. As a result, how does the authors ensure that the effect of Au-Col-BB NPs is homogenous for all the assays?
Answer: We agreed the consideration by reviewer. The polydispersity index (PDI) is an important parameter that describes the width or spread of the particle size distribution. Actually, the PDI value may vary from 0 to 1, where the colloidal particles with PDIs less than 0.1 implies monodisperse particles and the values more than 0.1 may imply polydisperse particle size distributions [Advances in Pharmaceutical Product Development and Research 2019, Pages 369-400]. PDI for Au, A-Col, and Au-Col-BB was approximately 0.43, 0.48, and 0.65, indicating they are all ideal particle size distribution. Besides, we have also included the new data of histogram into Figure 1C to further verify the dispersity of the AuNPs as well as description in the “Result” section “The DLS assay revealed that the incorporated BB would not affect the size distributions of Au-Col and Au-Col-BB” (Page 8, line 377-379), in the “Figure caption “The histogram of AuNP-derived nanocarrier were determined respectively using DLS.” (Page 9, line 408-409)
- Fig. 2A – Why is the 10 ug/mL Au-Col-BB treated cells having a higher viability compared to 1 ug/mL and 5 ugmL? Shouldn’t there be a dose dependent response? For Fig. 2B, BB only is able to give a greater cytotoxic effect compared to Au-Col-BB NPs – why is using Au-Col-BB NPs more advantageous for further applications?
Answer:
(1) We have carefully check out the data and corrected in the Figure 2A (A549). (Page 11)
(2) We agreed the valuable comment from reviewer. However, poor solubility either low bioavailability limit its therapeutic use or non-specific targeting for BB. According this study, the clathrin-mediated endocytosis seemed to be the favorite endocytic mechanism for the internalization of Au-Col-BB by Her-2 cells. Therefore, it was demonstrated that Au-Col-BB NPs explored and applied to target tumor cells precisely for avoiding any risk to harm normal cells or organs as well as to improve the quality and efficiency for targeting breast cancer treatments.
- For Fig. 3B, there is a lack of dose dependent response for the Au-Col-BB NPs – even with a concentration increase of 20 times (from 0.5 to 10 ug/mL), the cell viability is about the same. Is there any explanation for this? How does this inform the optimal concentration to use for subsequent assays? In Fig. 3C the labeling of Au-Col-BB is different than the other panels.
Answer:
(1) Thanks the valuable comment from the reviewer. We have carefully reproducible this data again and given the correct data into Figure 3B. (Page 12)
(2) The optimal concentration of Au-Col-BB was 1 ug/ml in the following experiments. We have included more detail description in each Figure caption [Figure 5 (Page 15, line 544); Figure 6 (Page 16, line 559); Figure 7 (Page 18, line 621)].
(3) We have corrected the labelling Fig. 3C. (Page 12)
- Based on the in vitroassays, it looks like BB already exerts a strong effect on killing cancer cells, comparable to Au-Col-BB NPs. However, this control group was not included in the mice work. Hence, it makes one wonder if the Au-Col-BB NPs indeed have better effects compared to BB, or is treatment with only BB enough (i.e., adding into drinking water, oral gavage etc)? It will be a stronger argument if there are other data supporting that either Au-Col-BB NPs are able to be more biocompatible (i.e., less inflammation, less toxicity to surrounding tissues than BB), or that Au-Col-BB NPs have a higher localization effect or longer circulation times than BB only.
Answer:
Thanks for the suggestion from the Reviewer. I have included data of BB only (20 mg) group to mice work (Figure 10). We have included the description in the “Results” section “Compared with the group of BB only (20mg), Au-Col--BB 20 mg obviously has better tumor suppressing ability (p<0.001) (Figure 10D), and prolongs the survival time of mice more effectively (p<0.001) (Figure 10 E).” (Page 21, line 678-681), in the “Figure caption” section “(A) Tumor volume of the experimental mice receiving various treatments is shown. Tumors were remarkably smaller in mice after treating with Au-Col-BB 20 mg compared to the mice treated with Control (PBS), Au-Col and BB only 20 mg. (B) Body weight curve of the experimental mice treating with various drugs is shown. The body weight was not remarkably changed in mice treated with different materials. (C) The images of tumors taken from the mice sacrificed at the end of the study. (D) Tumor weight of the experimental mice treating with various drugs is shown. Tumors were remarkably lighter in mice treated with the Au-Col-BB 20 mg than in mice treated Control (PBS), Au-Col and BB only 20 mg (*p<0.05, **p<0.01, ***p<0.001). (E) The survival median of various treatment mice groups (n=5) was evaluated up to day 70. The median survival of mice was significantly longer injected with Au-Col-BB 20 mg evaluated by Kaplan-Meier statistics and log-rank test.” (Page 21, line 684-693)
.
- In Fig. 10E, it seems that the survival time for Au-Col-BB 5 mg is even lesser than control group? Does this indicate any signs of toxicity that Au-Col-BB might be causing?
Answer: Thanks for the suggestion from the Reviewer. I have corrected the error on the drawing. The survival time for Au-Col-BB 5 mg is higher than control group (Figure 10) (Page 21)
- It is unclear from the material and methods section, as well as in the manuscript, how are the NPs being administered to the mice? It seems from Fig. 11 that the NPs are being treated via intratumor injection? Please provide clarity on this method in the appropriate sections and descriptions. Assuming it is indeed an intratumor injection, it is a highly localized injection at the tumor site. Hence, the significant effects in tumor volume reduction do not seem too surprising to me. One of the advantages of NPs is to improve blood circulation times and specificity of targeting to tumors due to the EPR effect. Hence, some common route of injections are intravenous and intraperitoneal. Why are these two routes not being used instead of intratumor? Will the Au-Col-BB NPs achieve the same extent of effect if it was done by either of these two routes?
Answer:
Thanks for the suggestion from the Reviewer. I have clearly described in the “Materials and Methods” section in Tumor xenograft mouse model “The 20-25g male BALB/c nude mice (approximately 2 months) were obtained from the National Laboratory Animal Center (Taiwan). Her-2 cells (2´106 ) in 50 μl matrigel were inoculated subcutaneously into the flanks of each mouse . Tumor growth was investigated through Vernier caliper and the tumor volume (V) was calculated based on the formula “V (mm3) = (D12 ´ D2)/2” (D1 and D2 represented as the shortest and longest tumor diameter). After the tumor was approximately 25±1mm3 (day zero), the following treatments were processed. All the tumor-bearing mice were randomly assigned into six groups: Au-Col, Au-Col-BB (5, 10, and 20 mg), control group (just treated with PBS), and BB only (20 mg). Each group included 6 mice (n=6). Then those formulas were given to mice on the 8th day, 11th day, and 14th day via the tail vein (intravenous) injection (50 μl). During the treatment period, the body weight and tumor size were measured every other day. After 30 days of treatment, mice were sacrificed through decapitation and tumors are removed and measured. In the survival time experiment, as in the previous experiment method, five mice in each group (n=5) were evaluated for survival time until the 70th day. The median survival of mice in all treatment group was calculated using Kaplan-Meier statistics and log-rank test.” (Page 7-8, line 347-363). The NPs were administered to the mice by tail vein (intravenous) injection.
Other minor comments:
- There are a lot of language errors including incoherent sentence structures, grammar mistakes and typos. For instance, a lot of sentences start with “Then the…” instead of using conjunctions such as “Next, we did this and that”; or “Additionally, we saw this and that”. The authors should improve on the language and proof-reading.
Answer:
Thanks for the suggestion from the Reviewer. We have modified the wording in the sentences and proofreading.
“Cancer cells trigger apoptosis resistance, metastasis, inflammation, and the breakdown of intercellular communication that cause poor immune response.” (Page 2, line 56-58)
“The Her-2 overexpressed breast cancer tends to grow faster and metastasize.” (Page 2, line 74-75)
“Various clinical therapeutic drugs such as Herceptin (traztuzumab) is effective to target Her-2.” (Page 2, line 75-76)
“…for cancer therapies owing to easiness of fabrication…” (Page 2, line 83-84)
“…nanoparticles-based therapies…” (Page 2, line 86)
“For instance, …” (Page 2, line 90)
“… serve as…” (Page 2, line 99)
“Based on the researches in cellular uptake mechanisms of nanoparticles, …” (Page 3, line 101)
“…technical advancements…” (Page 3, line 112)
“…due to lower cytotoxicity and decreasing…” (Page 3, line 119)
“BB is verified to…” (Page 3, line 125)
“…also inhibited…” (Page 3, line 131)
“…examination of cytotoxicity, investigation of cellular uptake mechanisms…” (Page 3, line 151-152)
“Breast cancer is the uncontrolled proliferation of breast cells that…” (Page 22, line 695)
- For the results section, there was very few data interpretation. The results are just described literally as how they are presented in the figures, without explaining clearly the rationale of doing the assays, and why the results make sense or are significant in proving a certain point/hypothesis.
Answer:
(1) Section 3.1.:
“Transmission electron microscope (TEM) was further applied to observe material surface of Au (a), Au-Col (b), and (c) Au-Col-BB, demonstrating the spherical shape of nanoparticles (Figure 1B).” (Page 8, line 375-377)
“The UV-Vis absorption peak at 520 nm was observed for the gold-containing nanoparticles, Au, Au-Col, and Au-Col-BB, indicating the presence of Au in each sample (Figure 1E).” (Page 8, line 391-392)
(2) Section 3.2.:
“The above results indicated Au-Col-BB could significantly inhibit cell proliferation and induce cell apoptosis in A549 lung cancer cell and Colo-205 human colon cancer cell lines. Further, Her-2 breast cancer cells were treated with Au-Col-BB and BB (alone) to evaluate the cytotoxicity. We found that Au-Col-BB also had the potent for inhibiting Her-2 cell growth and increasing the amount of apoptotic and death cells, verifying the efficiency of cytotoxicity in breast cancer cells.” (Page 10, line 443-448)
(3) Section 3.3.:
“To investigate the biocompatibility of Au-Col-BB between normal and cancer cells, non-transformed BAEC were chosen to compare with Her-2 breast cancer cells.” (Page 10, line 450-451)
“The above evidence showed Au-Col-BB would increase cell viability of non-transformed BAEC, but significantly induce cell apoptosis in Her-2 cells, supporting that Au-Col-BB had specific cytotoxicity effect.” (Page 10-11, line 467-469)
(4) Section 3.4.:
“The results elucidated after the nanocarrier Au-Col-BB could be significantly uptake by both BAEC and Her-2 cells compared to Au-Col.” (Page 14, line 514-516)
(5) Section 3.5.:
“Based on the results of Lysotracker assay, we observed that Au-Col-BB would not degrade in lysosome after the autophagy and was seen to be stable, supporting the high drug delivery capacity of Au-Col carrying BB.” (Page 14, line 534-536)
“According to the above results, we investigated the mechanism of endocytotic route in both BAEC and Her-2 cell line and found that the uptake of Au-Col-BB nanocarrier could be significantly inhibited by CPZ and Baf lysosomal inhibitors due to the size of Au-Col-BB nanocarrier (227 nm), while CPZ inhibited clathrin-mediated endocytosis and Baf interfered cell autophagy.” (Page 17, line 587-591)
(6) Section 3.6.:
“The above evidence demonstrated after treating with Au-Col-BB, the MMP-9 activity in Her-2 cell line was significantly suppressed, leading to the less migration distance. But no influence in non-transformed BAEC cell, indicating the specific effect of Au-Col-BB.” (Page 17, line 603-606)
(7) Section 3.7.:
“The above evidence indicated Au-Col-BB could induce Her-2 cell apoptosis through inducing the expression of Bax and p21, and also decreasing the expression of anti-apoptotic proteins, Bcl-2 and Cyclin D1. Supporting that Au-Col-BB nanocarrier has the capacity of anti-cancer.” (Page 19, line 647-650)
(8) Section 3.8.:
“Compared with the group of BB only (20mg), Au-Col--BB 20 mg obviously has better tumor suppressing ability (p<0.001) (Figure 10D), and prolongs the survival time of mice more effectively (p<0.001) (Figure 10E).” (Page 21, line 678-681)
- 6 should be LysoTracker dye, not Lysotracer dye. Fig. 6D labeling is missing.
Answer:
We have modified “Lysotracer” to “Lysotracker” and we also labeled the Figure 6D. (Page 16)
- 11 resolution is fuzzy.
Answer:
We have modified and make it to be more clear in Figure 11 and moved this figure into “Graph Abstract” by reviewer’s suggestion.
- The statistics labeling for the figures are confusing. For example, in Fig. 3A and 3B, it is not clear what are the two treatment groups that are being compared and which * corresponds to which comparison. Please indicate the statistics clearly and consistently for each figure.
Answer:
We have modified the statistics labeling in Figure 3, 4, 5, 6, and 7 to make them more clearly for the readers. We also included more detail description “compared to the control group” in several Figure captions. [Figure 2 (Page 11, line 477-478), Figure 3 (Page 12, line 486), Figure 4 (Page 14, line 494), Figure 5 (Page 15, line 544-545), Figure 6 (Page 16, line 560), Figure 7 (Page 18, line 622), Figure 8 (Page 19, line 635)]
Reviewer 3 Report
In this paper the authors developed a novel cancer treatment by combining AuNP with collagen and berberine to create an Au-Col-BB complex. I recommend the publication of the paper after minor reviews are done.
- Please, make the abstract more direct and fluid
- On the intro (page 2, line 52-60). I suggest the author to focus it on breast cancer, because that was the model used on the paper.
- Page 2, line 60. What are the bottlenecks that need to be conquered? Please, add examples and citations.
- In the methods (page 3-4, line 148-159), the authors say twice “Described by our previous report”, I suggest removing the first one and leave the citation at the end of the paragraph.
- In the Au-Col-BB preparation, are the nanoparticles colloidal? Do the need a reduction agent such as citrate to synthesize it?
- In the material characterization (page 4, line 163). The UV-vis typical absorption peak for Au is 520nm, however if AuNPs are aggregated the peak can shift to values above 520nm, and looking at figure 1, how the authors explain the enhanced aggregation on the insert B(b), and (c). I suggest to built a histogram showing the dispersity of the AuNPs size, in order to verify if they are monodispersed or polydisperse.
- Why the authors used A549, Colo-205, and BAEC cells if the focus were on breast cancer. Please, add more details.
- Please, change the figure 11 to be the figure 1 or the graphical abstract.
Author Response
Reviewer 3:
Comments and Suggestions for Authors
In this paper the authors developed a novel cancer treatment by combining AuNP with collagen and berberine to create an Au-Col-BB complex. I recommend the publication of the paper after minor reviews are done.
- Please, make the abstract more direct and fluid.
Answer:
Thanks the valuable from reviewer. We have modified the wording as suggested by reviewer in the “Abstract” section.
“Gold nanoparticles (AuNPs) were fabricated with biocompatible collagen (Col) and then conjugated with berberine (BB), denoted as Au-Col-BB, to investigate the endocytic mechanisms in Her-2 breast cancer cell line and in bovine aortic endothelial cells (BAEC). Owing to the superior biocompatibility, tunable physicochemical properties, and potential functionalization with biomolecules, AuNPs have been well studied as carriers of biomolecules for diseases and cancer therapeutics. Composites of AuNPs with biopolymer such as fibronectin or Col have been revealed to increase cell proliferation, migration, and differentiation. BB is a natural compound with impressive health benefits such as lowering blood sugar and reducing weight. Besides, BB can inhibit cell proliferation by modulating cell cycle progress and autophagy, and induce cell apoptosis in vivo and in vitro. In the current research, BB was conjugated on the Col-AuNP composite (“Au-Col”). The UV-Visible spectroscopy and infrared spectroscopy confirmed the conjugation of BB on Au-Col. The particle size of the Au-Col-BB conjugate was about 227 nm, determined by dynamic light scattering. Furthermore, Au-Col-BB was less cytotoxic to BAEC vs. Her-2 cell line in terms of MTT assay and cell cycle behavior. Au-Col-BB, compared to Au-Col, showed greater cell uptake capacity and potential cellular transportation by BAEC and Her-2 using the fluorescence-conjugated Au-Col-BB. Besides, the clathrin-mediated endocytosis and cell autophagy seemed to be the favorite endocytic mechanism for the internalization of Au-Col-BB by BAEC and Her-2. Au-Col-BB significantly inhibited cell migration in Her-2, but not in BAEC. Moreover, apoptotic cascade proteins such as Bax and p21 were expressed in Her-2 after the treatment of Au-Col-BB. The tumor suppression was examined in a model of xenograft mice treated with Au-Col-BB nanovehicles. Results demonstrated that the tumor weight was remarkably reduced by the treatment of Au-Col-BB. Altogether, the promising findings of Au-Col-BB nanocarrier on Her-2 breast cancer cell line suggest that Au-Col-BB may be a good candidate of anticancer drug for the treatment of human breast cancer.” (Page 1-2, line 30-52)
- On the intro (page 2, line 52-60). I suggest the author to focus it on breast cancer, because that was the model used on the paper.
Answer:
Thanks for the suggestion from the Reviewer. We have included the references to focus on the breast cancer. In the “Introduction” section “Although chemotherapy is an effective procedure to decrease the volume of primary tumor before surgery, a long-term chemotherapy treatment can cause various side effects such as nausea, vomiting, fatigue, hair loss, leukopenia, mucositis, neurosensory disorders, and taste alterations in cancer patients and induce multidrug resistance [1-4].” (Page 2, line 59-63)
References:
[2] Wind, N.; Holen, I. Multidrug resistance in breast cancer: from in vitro models to clinical studies. International journal of breast cancer 2011, 2011, doi: 10.4061/2011/967419.
[3] Binkley, J.M.; Harris, S.R.; Levangie, P.K.; Pearl, M.; Guglielmino, J.; Kraus, V.; Rowden, D. Patient perspectives on breast cancer treatment side effects and the prospective surveillance model for physical rehabilitation for women with breast cancer. Cancer 2012, 118, 2207-2216.
[4] Shapiro, C.L.; Recht, A. Side effects of adjuvant treatment of breast cancer. New England Journal of Medicine 2001, 344, 1997-2008.
- Page 2, line 60. What are the bottlenecks that need to be conquered? Please, add examples and citations.
Answer:
We have included samples and citation in the “Introduction” section “In the same way, there are still various bottlenecks that need to be conquered urgently. For instance, new drug development for cancer diseases requires plenty of time and costs expensive [5]. A literature figured out “Drug repurposing” to be a clinical strategy through using of approved drugs for breast cancer therapeutics [5] such as the combination of insulin like growth factor 1 receptor (IGF1R) inhibitors with approved drug Rapamycin [6] demonstrates the inhibition of breast cancer cells proliferation [7].” (Page 2, line 63-69)
Reference:
[5] Zhang, Z.; Zhou, L.; Xie, N.; Nice, E.C.; Zhang, T.; Cui, Y.; Huang, C. Overcoming cancer therapeutic bottleneck by drug repurposing. Signal transduction and targeted therapy 2020, 5, 1-25.
[6] Benjamin, D.; Colombi, M.; Moroni, C.; Hall, M.N. Rapamycin passes the torch: a new generation of mTOR inhibitors. Nature reviews Drug discovery 2011, 10, 868-880.
[7] Rugo, H.S.; Trédan, O.; Ro, J.; Morales, S.M.; Campone, M.; Musolino, A.; Afonso, N.; Ferreira, M.; Park, K.H.; Cortes, J. A randomized phase II trial of ridaforolimus, dalotuzumab, and exemestane compared with ridaforolimus and exemestane in patients with advanced breast cancer. Breast cancer research and treatment 2017, 165, 601-609.
- In the methods (page 3-4, line 148-159), the authors say twice “Described by our previous report”, I suggest removing the first one and leave the citation at the end of the paragraph.
Answer:
Thanks for the valuable suggestion from the Reviewer. We have removed the first one of “Described by our previous report” in section 2.1. and leave the citation at the end of the paragraph. (Page 4, line 171)
- In the Au-Col-BB preparation, are the nanoparticles colloidal? Do the need a reduction agent such as citrate to synthesize it?
Answer:
Thanks the valuable comment from the reviewer. We have included more detail description in the “Materials and Methods” section “The physical gold nanoparticles were purchased from Gold NanoTech, Inc (GNT, Taiwan). The GNT Gold is 99.99% pure, manufactured by physical vapor deposition (PVD) processing, which is different from chemical synthesis used by commercially available nanogold products. GNT Gold contains no other heavy metals or toxic compounds.” (Page 4, line 157-160)
- In the material characterization (page 4, line 163). The UV-vis typical absorption peak for Au is 520nm, however if AuNPs are aggregated the peak can shift to values above 520nm, and looking at figure 1, how the authors explain the enhanced aggregation on the insert B(b), and (c). I suggest to built a histogram showing the dispersity of the AuNPs size, in order to verify if they are monodispersed or polydisperse.
Answer:
Thanks for the suggestion from the Reviewer. We have also included the new data of histogram into Figure 1C to further verify the dispersity of the AuNPs as well as description in the “Result” section “The DLS assay revealed that the incorporated BB would not affect the size distributions of Au-Col and Au-Col-BB” (Page 8, line 377-379), in the “Figure caption “The histogram of AuNP-derived nanocarrier were determined respectively using DLS” (Page 9, line 408-409)
- Why the authors used A549, Colo-205, and BAEC cells if the focus were on breast cancer. Please, add more details.
Answer:
(1) We want to evaluate the potent of Au-Col-BB for various tumor cancer treatments simultaneously.
(2) Angiogenesis occurs during neoplasia, that leads to the proliferation of endothelial cells. We want to realize the efficacy of Au-Col-BB clinical treatment in comparison with normal cells (BAEC) and breast cancer cells (Her-2) whether Au-Col-BB can reduce the standard effects, increase the efficiency of specific therapeutic effects and decrease the cytotoxicity of normal cells
- Please, change the figure 11 to be the figure 1 or the graphical abstract.
Answer:
We have moved the Figure 11 into “Graph Abstract” by reviewer’s suggestion.
Round 2
Reviewer 1 Report
The suggestions/comments are given below.
- Authors only correct the errors that I pointed out. Please check all the description in the manuscript carefully. There are still a lot of mislabeling and description. Here, I just indicated some errors. For example, Line 595-596, what do you mean “… quantification results of Her-2 and Her-2 indicated that the gelatin degradation by matrix metalloproteinase 2.”; Line 595-596, the authors indicated that ”BAEC (Figure 8E) and Her-2 (Figure 8F) demonstrated that the migration ability was observed to be significantly decreased after treatment particularly in Her-2 breast cancer cells (Figure 8H) for Au-Col- BB (10 μg/ml) compared to BAEC (Figure 8G).”, but the data indicated not only 10 μg/ml of Au-Col- BB can reduced Her-2 cells migration, but all the concentrations of Au-Col- BB can significantly reduce Her-2 cells migration. The authors did not correct the description of the legend’s description in Figure 9A&C. it should be Western blot, not zymogram test (Line 652, Line 660).
- Authors need to check the Statistics analysis is correct or not. For example, Figure 6B, it may me confuse the statistically result at 2 hr. A similar situation in Figure 6C, E&F, and Fig 4 C&D; in Fig 10D, please use different symbols to mark statistically difference to different control.
Author Response
Comments and Suggestions for Authors
The suggestions/comments are given below.
Authors only correct the errors that I pointed out. Please check all the description in the manuscript carefully. There are still a lot of mislabeling and description. Here, I just indicated some errors.
- For example, Line 595-596, what do you mean “… quantification results of Her-2 and Her-2 indicated that the gelatin degradation by matrix metalloproteinase.”;
Answer:
We have deleted the wrong sentence and rewritten appropriate sentences. “The zymography images of BAEC and Her-2 were shown as Figure 8A&8B. Based on the quantitative results in Figure 8C for BAEC, there was no significant difference in both MMP-2/9 compared to the control at 48 hr. In Figure 8D for Her-2 cell line, the results indicated that the expression of MMP-9 were the lowest in Au-Col-BB (10 μg/ml) at 48 hr. And the expression of MMP-2 was significantly decreased in BB (1 μg/ml), Au-Col-BB (5, 10 μg/ml) groups.” (Page 17, Line 603-608)
- Line 595-596, the authors indicated that” BAEC (Figure 8E) and Her-2 (Figure 8F) demonstrated that the migration ability was observed to be significantly decreased after treatment particularly in Her-2 breast cancer cells (Figure 8H) for Au-Col- BB (10 μg/ml) compared to BAEC (Figure 8G).”, but the data indicated not only 10 μg/ml of Au-Col- BB can reduced Her-2 cells migration, but all the concentrations of Au-Col- BB can significantly reduce Her-2 cells migration.
Answer:
We have rewritten for appropriate sentences. “Furthermore, BAEC (Figure 8E) and Her-2 (Figure 8F) demonstrated the cell the migration ability influenced by each treatment. In Figure 8G, the results indicated that each treatment did not inhibit BAEC migration, but promoted the migration distance at 24 and 48 hr. On the contrary, the migration distance of Her-2 cells was significantly decreased in each treatment. However, after treated with Au-Col-BB (10 μg/ml) for 48 hr, the migration distance was 0 μm. The above evidence demonstrated after treating with Au-Col-BB (10 μg/ml), the MMP-9 activity in Her-2 cell line was significantly suppressed, leading to the lowest migration distance (Figure 8H)” (Page 17, Line 609-616)
- The authors did not correct the description of the legend’s description in Figure 9A&C. it should be Western blot, not zymogram test (Line 652, Line 660).
Answer:
Thank you for the comment from the Reviewer. We have changed the description in Figure 9A “The representative western blot images of BAEC were demonstrated.” (Page 20, Line 663) and in Figure 9C “The representative western blot images of Her-2 cells were shown.” (Page 20, Line 669)
- Authors need to check the Statistics analysis is correct or not. For example, Figure 6B, it may me confuse the statistically result at 2 hr. A similar situation in Figure 6C, E&F, and Fig 4 C&D; in Fig 10D, please use different symbols to mark statistically difference to different control.
Answer:
Thanks for the valuable suggestion from the Reviewer. We have corrected the statistically result in Figure 6B, C, E & F and used different symbols in Figure 10D.
We have checked the manuscript and modified the description and statistical labeling in Figures and Result section (blue mark):
(1) “The DLS assay revealed the size distribution intensity of AuNPs, Au-Col, Au-Col-BB, and Au-Col-BB-FITC (Figure 1C).” (Page 8, Line 377-379)
(2) “(C) The size distribution intensity of AuNP-derived nanocarrier were determined by using DLS.” (Page 9, Line 408-409)
(3) “(E) UV-Visible spectra confirmed each nanomaterial containing AuNPs with the typical absorption peak at 520 nm.” (Page 9, Line 414)
(4) “By MTT assay, the effects of Au-Col-BB on cytotoxicity were investigated with A549, Colo-205 and Her-2 cancer cell lines.” (Page 10, Line 418-419)
(5) “Figure 3A-B demonstrated that BB and Au-Col-BB could induced a significant cytotoxicity on Her-2 cell line, particularly in Au-Col-BB (10 μg/ml) group at 24, 48 and 72 hr.” (Page 10, Line 455-456)
(6) “For non-transformed BAEC, the sub-G1 population was slightly increased after treating with BB (1 μg/ml) and Au-Col-BB (0.5, 1, 5, and 10 μg/ml) compared to the control (Figure 3C). To view cell cycle influence, Her-2 cancer cells treated with Au-Col-BB especially for 10 μg/ml demonstrated the greatest sub-G1 cell populations (p< 0.01) compared to the control (Figure 3D). Further, Figure S1 demonstrated the cell viability of both BAEC and Her-2 cell line culturing with various concentrations of BB (0.5, 1, 5, and 10 μg/ml), the results indicated treating with BB 10 μg/ml had the lowest cell viability compared to other groups. The population of apoptotic cells for BAEC and Her-2 cells treating with various materials was evaluated by Annexin V-PI double staining assay. The images of flow cytometry were displayed as Figure 4A (BAEC) & 4B (Her-2). In Figure 4D, the population of apoptotic (Annexin-V positive) Her-2 cells was 80.6 %, which was the highest induced by Au-Col-BB (10 μg/ml). And the population of viable Her-2 cells was the lowest (18.2 %) after treating with Au-Col-BB (10 μg/ml). On the contrary, the quantitative results in Figure 4C indicated that after BAEC treating with Au-Col-BB (10 μg/ml), the population of apoptotic cells (Annexin-V positive) was 0.78 %, and the population of viable cells was 95.63 %. The above evidence showed Au-Col-BB (10 μg/ml) could efficiently induce cell apoptosis in Her-2 cells, supporting that Au-Col-BB had specific cytotoxicity effect.” (Page 10-11, Line 458-474)
(7) “The cell growth was observed with significantly inhibition of each cell line especially for Au-Col-BB (10 μg/ml) and BB (1 μg/ml) at 48 hr.” (Page 11-12, Line 476-477)
(8) “The cell viability was measured at 24, 48, and 72 hr. Cell growth of BAEC was increased at 48 and 72 hr seeded on different materials. In contrast, the growth of Her-2 cell line was significantly inhibited particularly in Au-Col-BB (10 μg/ml) group at 24, 48 and 72 hr.” (Page 12, Line 485-487)
(9) “G0/G1 population of BAEC was significantly reduced in BB group (1 μg/ml), but slightly decreased in Au-Col-BB groups.” (Page 12, Line 488-490)
(10) “Cell populations of Her-2 cell line at sub-G1 phase was obviously increased especially in Au-Col-BB (10 μg/ml) group. And the population of G0/G1 phased was significantly decreased after various treatments.” (Page 12-13, Line 490-492)
(11) “Figure 4. Annexin V-PI double staining to detect apoptotic BAEC and Her-2 cells by using flowcytometry at 48 hr. The cell population of (A) BAEC and (B) Her-2 cells was demonstrated. (C) BAEC appeared to be not injured by Au-Col-BB in terms of apoptotic and dead cell amount. However, while incubating with BB (1 μg/ml), the amount of Annexin-V positive cell and PI positive cell was 1.64 % and 6.19 %, and the dead cell amount was figured out for 6.99 %. The quantification of viable BAEC cells also showed high survival ratio in each Au-Col-BB group, but slightly increased in BB (1 μg/ml). (D) The quantitative results evaluated that the amount of apoptotic Her-2 cells (Annexin-V positive) was remarkably increased especially in Au-Col-BB (10 μg/ml), which was figured out for 80.6 %. The amounts of dead cells induced by BB (1 μg/ml) was 6.15 % that significantly more than control and other groups. The quantification of viable Her-2 cells also showed lowest survival ratio was 18.2 % in Au-Col-BB (10 μg/ml) group. *p < 0.05, **p < 0.01: compared to the control group.” (Page 13-14, Line 493-501)
(12) “…the uptake amount of Au-Col and Au-Col-BB was increased to ~ 4.01 and ~ 4.96 fold at 2 hr, and achieved to ~ 7.11 and ~ 9.17 fold at 24 hr, compared to the control group.” (Page 14, Line 509-511)
(13) “Further, the uptake amount in Her-2 cell line was also measured. In IF, the uptake amount for Au-Col was ~ 4.2 fold at 2 hr, ~ 5.72 fold at 24 hr; and for Au-Col-BB was ~4.67 fold at 2 hr and ~6.15 fold at 24 hr (Figure 5E). However, the FACS results indicated the uptake amount of Au-Col and Au-Col-BB was both significantly higher at each time point compared to the control. For Au-Col was ~ 2.1 and ~2.2 fold at 2 and 24 hr. For Au-Col-BB group was ~ 2 fold at 2 hr, and ~ 2.1 fold at 24 hr, respectively (Figure 5F). The images of uptake efficiency at 30 min and 24 hr was also evaluated by using non-transformed BAEC (Figure S2A, 2B) and Her-2 breast cancer cells (Figure S2C, S2D) which was treated with FITC conjugated Au-Col and Au-Col-BB. The results elucidated after the nanocarrier Au-Col-BB could be significantly uptake by both BAEC and Her-2 cells compared to Au-Col.” (Page 14, Line 513-523)
(14) “…increased intensity of Au-Col and Au-Col-BB at both 2 hr (~ 4.48 fold and ~ 4.92 fold) and 24 hr (~ 5.2 fold and ~ 5.28 fold) (Figure 6C). Furthermore, the FITC intensity for Her-2 cell line was also investigated. The uptake amounts in Her-2 at 30 min quantified by IF method was ~ 4.67 fold (Au-Col) and ~ 6.41 fold (Au-Col-BB). However, the uptake amount of Au-Col-BB was decreased to ~ 4.84 fold and Au-Col was ~ 5.04 fold at 2 hr. Further, at 24 hr, the intensity of Au-Col increased to ~ 6.76 fold, and the intensity of Au-Col-BB was slightly decreased to ~ 4.69 fold (Figure 6E). In contrast, the FACS method indicated different results that the intensity of Au-Col-BB in Her-2 cells (~ 4.98 fold) were greater than Au-Col (~ 4.65 fold) at 2 hr. The similar trend was also observed at 24 hr (Au-Col-BB: ~ 5.25 fold, Au-Col: ~4.98 fold) (Figure 6F). Additionally, the images of FITC fluorescence intensity were also demonstrated at 30 min and 24 hr by using non-transformed BAEC (Figure S3A&S3B) and Her-2 breast cancer cell line (Figure S3C&S3D)…” (Page 14, Line 532-544)
(15) “Figure 5. Assessment of cell uptake ability in BAEC and Her-2 cell line. Au-Col and Au-Col-BB were firstly conjugated with fluorescent dye (FITC) to investigate inside cell transportation, then observed by using fluorescent microscopy. The immunofluorescence (IF) images at 2 hr were displayed as (A) BAEC and (D) Her-2 cells. The semi-quantified uptake ability of BAEC was demonstrated through (B) IF and (C) fluorescence activated cell sorter (FACS) method. The results from IF and FACS both indicated the uptake amount of Au-Col-BB was significantly increased at 2 and 24 hr. Besides, in Her-2 breast cancer cell line, the uptake efficiency was also quantified based on (E) IF and (F) FACS method. The IF results also indicated the uptake amount of Au-Col-BB was the greatest in Her-2 cells at 2 and 24 hr. The quantified data from FACS showed the uptake amount of Au-Col-BB and Au-Col was higher than the control at each time point. The concentration of Au-Col-BB was 1 ug/ml for the experiments. Scale bars = 20 μm. *p < 0.05, **p < 0.01: compared to the control group.” (Page 15, Line 548-556)
(16) “The zymography images of BAEC and Her-2 were shown as Figure 8A&8B. Based on the quantitative results in Figure 8C for BAEC, there was no significant difference in both MMP-2/9 compared to the control at 48 hr. In Figure 8D for Her-2 cell line, the results indicated that the expression of MMP-9 were the lowest in Au-Col-BB (10 μg/ml) at 48 hr. And the expression of MMP-2 was significantly decreased in BB (1 μg/ml), Au-Col-BB (5, 10 μg/ml) groups. Furthermore, BAEC (Figure 8E) and Her-2 (Figure 8F) demonstrated the cell the migration ability influenced by each treatment. In Figure 8G, the results indicated that each treatment did not inhibit BAEC migration, but promoted the migration distance at 24 and 48 hr. On the contrary, the migration distance of Her-2 cells was significantly decreased in each treatment. However, after treated with Au-Col-BB (10 μg/ml) for 48 hr, the migration distance was 0 μm. The above evidence demonstrated after treating with Au-Col-BB (10 μg/ml), the MMP-9 activity in Her-2 cell line was significantly suppressed, leading to the lowest migration distance (Figure 8H).” (Figure 17, Line 603-616)
(17) “In Her-2 cell line, matrix metalloproteinases MMP-9 were significantly reduced after treated with Au-Col-BB, especially at the concentration of 10 μg/ml.” (Page 19, Line 637-638)
(18) “BAEC did not express more apoptotic cascade proteins such as Bax and p21 after culturing with Au-Col-BB (0.5, 1, 5, 10 μg/ml), but was found to significantly expressed after treated with BB (1 μg/ml), which were increased to ~ 4.5, and ~ 7.8 fold, respectively (Figure 9A-B).” (Page 19, Line 648-651)

Reviewer 2 Report
Authors have addressed the comments.
Author Response
Comments and Suggestions for Authors
Authors have addressed the comments.
Answer:
Thanks for the valuable suggestions and comments from the Reviewer.

Round 3
Reviewer 1 Report
The authors have addressed all of my comments and concerns in the revised version. I have no additional comments.